# HCOOH distributions from IASI for 2008-2014: comparison with ground-based FTIR measurements and a global chemistry-transport model

M. Pommier[1,*], C. Clerbaux[1,2], P.-F. Coheur[2], E. Mahieu[3], J.-F. Müller[4], C. Paton-Walsh[5], T. Stavrakou[4], C. Vigouroux[4]

1 LATMOS/IPSL, UVSQ Université Paris-Saclay, UPMC Univ. Paris 06, CNRS, Guyancourt, FranceFrance
2 Spectroscopie de l'Atmosphère, Chimie Quantique et Photophysique, Université Libre de Bruxelles (ULB), Brussels, Belgium
3 Institute of Astrophysics and Geophysics of the University of Liège, Liège, Belgium
4 Royal Belgian Institute for Space Aeronomy (BIRA-IASB), Avenue Circulaire 3, 1180 Brussels, Belgium
5 School of Chemistry, University of Wollongong, Wollongong, Australia
* Now at Norwegian Meteorological Institute, Oslo, Norway

Correspondence to: M. Pommier (matthieup@met.no)

**Abstract.**

Formic acid (HCOOH) is one of the most abundant volatile organic compounds in the atmosphere. It is a major contributor to rain acidity in remote areas. There are, however, large uncertainties on the sources and sinks of HCOOH and therefore HCOOH is misrepresented by global chemistry-transport models. This work presents global distributions from 2008 to 2014 as derived from the measurements of the Infrared Atmospheric Sounding Interferometer (IASI), based on conversion factors between brightness temperature differences and representative retrieved total columns over seven regions: Africa N, Africa S, Amazonia, Atlantic, Australia, Pacific and Russia. The dependence of the measured HCOOH signal to the thermal contrast is taken into account in the conversion method. This conversion presents errors lower than 20% for total columns ranging between 0.5 and $1\times10^{16}$ molec/cm$^2$ but reaches higher values, up to 78%, for columns that are lower than $0.3\times10^{16}$ molec/cm$^2$. Signatures from biomass burning events are highlighted, such as in the Southern Hemisphere and in Russia, as well as biogenic emission sources, e.g. over the Eastern US. A comparison between 2008 and 2014 with ground-based FTIR measurements obtained at four locations (Maido and Saint-Denis at La Réunion, Jungfraujoch and Wollongong) is shown. Although IASI columns are found to correlate well with FTIR data, a large bias (>100%) is found over the two sites at La Réunion. A better agreement is found at Wollongong with a negligible bias. The comparison also highlights the difficulty of retrieving total columns from IASI measurements over mountainous regions such as Jungfraujoch. A comparison of the retrieved columns with the global chemistry-transport model IMAGESv2 is also presented, showing good representation of the seasonal and inter-annual cycles over America, Australia, Asia and Siberia. A global model underestimation of the distribution and a misrepresentation of the seasonal cycle over India are also found. A small positive trend in the IASI columns is observed over Australia, Amazonia and India over the 2008-2014 period (from 0.7 to 1.5%/year), while a decrease of ~0.8%/year is measured over Siberia.

## 1. Introduction

Formic acid (HCOOH) is among the most abundant volatile organic compounds (VOCs) present in the atmosphere. Along with acetic acid it is a major contributor to the acidity of precipitation, especially in remote regions (Keene and Galloway, 1988; Andreae et al., 1988). HCOOH has small direct emissions from vegetation (Keene and Galloway, 1984, 1988), ants (Graedel and Eisner, 1988), biomass burning (e.g. Goode et al., 2000), soils (Sanhueza and Andreae, 1991), agriculture (e.g., Ngwabie et al., 2008), and motor vehicles (Kawamura et al., 1985; Grosjean, 1989), but is mainly a secondary product from

other organic precursors. The largest global chemical sources of HCOOH include isoprene, monoterpenes, other terminal alkenes (e.g., Neeb et al., 1997; Lee et al., 2006; Paulot et al., 2011), alkynes (Hatakeyama et al., 1986; Bohn et al., 1996), and acetaldehyde (Andrews et al., 2012; Clubb et al., 2012). HCOOH is mainly removed from the troposphere through wet and dry deposition, and to a lesser extent through oxidation by the OH radical.

HCOOH is a short-lived species and its lifetime is mainly determined by the precipitation rate. The lifetime ranges between

2 days during the rainy season and 6 days in the dry season in the boundary layer (Sanhueza et al., 1996). The global lifetime in the troposphere is 3–4 days (Paulot et al., 2011; Stavrakou et al., 2012). Photochemical loss is relatively slow ($\tau \sim 25$ days), so that any HCOOH formed or vented outside of the boundary layer can be transported over long distances in the free troposphere (Paulot et al., 2011).

Our knowledge about sources and sinks of HCOOH is still incomplete despite the numerous studies that have been published

during the last decade. In current emissions inventories, such as the biogenic emission inventory MEGAN-MACC (Sindelarova et al., 2014), the main source regions are located in tropical regions as presented in Fig. 1 for the period between 2008 and 2010. A recent work shows a possible source of HCOOH over the Arctic Ocean (Jones et al., 2014). The study by Stavrakou et al. (2012) highlights a misrepresentation of emissions from tropical and boreal forests in models compared to total columns retrieved from Infrared Atmospheric Sounding Interferometer (IASI) observations by Razavi et

al. (2011). Millet et al. (2015), Paulot et al. (2011) as well as Stavrakou et al. (2012) point to the existence of one or more large missing sources. These studies suggest an important gap in our current understanding of hydrocarbon oxidation and/or the existence of unknown direct fluxes of HCOOH.

Nadir looking atmospheric sensors allow to derive global distributions for trace gases, with a limited vertical sensitivity as compared to airborne or ground-based measurements. Their extended spatial coverage allows observing remote regions

which are sparsely studied by field campaigns. Only a few satellites provide tropospheric HCOOH observations, such as the nadir-viewing instrument IASI (e.g. Razavi et al., 2011) and the Tropospheric Emission Spectrometer (TES) (e.g. Cady-Pereira et al., 2014). The Michelson Interferometer for Passive Atmospheric Sounding (MIPAS) limb instrument provided monthly global distributions of HCOOH around 10 km (Grutter et al., 2010), and the solar-occultation Atmospheric Chemistry Experiment (ACE) provides seasonal global distribution in the upper troposphere (e.g. González Abad, 2009).

The data used in this study are provided by IASI. This instrument has two important advantages: a low radiometric noise and a high spatial coverage. HCOOH is a weak absorber, so it is a challenge to retrieve total columns from the IASI radiance. Global distributions of HCOOH over land were initially derived using a method based on brightness temperature difference and using forward simulations (Razavi et al., 2011). More recently, R'Honi et al. (2013) developed a specific method to study extreme events occurring during the large fires in Russia during the summer 2010. These studies however highlighted

discrepancies between the retrieved distributions and especially within enriched HCOOH air masses as, for instance, over large forest fires. Indeed, the total columns from R'Honi et al. (2013) were on average a factor of 2 lower than in Razavi et al. (2011) (around a factor of 1.5 for columns higher than $5\times10^{16}$ molec.cm$^{-2}$ and 2.3 for columns lower than $5\times10^{16}$ molec.cm$^{-2}$). In this paper, we present an update of the method used in Razavi et al. (2011), in order to derive HCOOH distributions over both land and sea, suitable for both enhanced and background concentrations over the period 2008-2014.

Section 2 introduces the IASI mission and explains the methodology used, based on an optimal estimation method (OEM) retrieval over selected areas, used to design a fast retrieval methodology based on the brightness temperature difference and conversion factors. In Section 3, global distributions are shown; the products are compared to ground-based measurements and to the global chemistry-transport model (CTM) IMAGESv2, providing an analysis of the seasonal and inter-annual variability of HCOOH columns. The conclusions are given in Section 4.


## 2. IASI HCOOH columns

### 2.1. The IASI mission

IASI is a nadir-viewing Fourier transform spectrometer instrument. Currently two instruments are in orbit. The first model was launched on board the METOP-A platform in October 2006 providing now more than eight years of observations. The

second instrument was launched in September 2012. Owing to its wide swath, each instrument delivers near global coverage twice per day at around 9:30 local time (AM and PM). IASI measures in the thermal infrared part of the spectrum, between 645 and 2760 cm$^{-1}$. It records radiance from the Earth's surface and the atmosphere with an apodized spectral resolution of 0.5 cm$^{-1}$, spectrally sampled at 0.25 cm$^{-1}$. IASI has a good radiometric performance, around 0.15 K (or around $2\times10^{-6}$ W cm$^{-2}$ sr$^{-1}$ / cm$^{-1}$) in the HCOOH spectral range (~1105 cm$^{-1}$) for a reference blackbody at 280 K (Clerbaux et al., 2009).

Analysis of the mean of the normalized Jacobians (Fig. 2) over the spectral range used by IASI for the HCOOH retrievals (1095-1114 cm$^{-1}$), for a set of representative geographical regions (see Fig. 1 and next section), shows that IASI is sensitive to tropospheric HCOOH between 1 and 6 km. These Jacobians represent the sensitivity of IASI and the radiative transfer model to the abundance of HCOOH in a fixed atmosphere. This corresponds to the mean of the Jacobians simulated from the selected spectra over the studied regions.


### 2.2 Retrieval approach

Processing seven years (2008-2014) of IASI data using an iterative method such as the OEM (Rodgers, 2000) is computationally demanding. Hence we have chosen a fast approach based on brightness temperature differences ($\Delta T_b$) between spectral channels with and without the signature of the target gas, to extract information without performing a full retrieval. In a second step, the $\Delta T_b$ were converted into total columns of HCOOH using conversion factors derived from a set of data retrieved by OEM. This approach was adapted from the previous works for other IASI weak absorbers, such as methanol (Razavi et al. 2011), sulfur dioxide (Clarisse et al., 2008) and ammonia (Van Damme et al., 2014).

In the current work, the main difference with the previous IASI HCOOH determination in Razavi et al. (2011) is the use of retrieved total columns over selected regions to determine conversion factors, instead of the use of forward simulations. The OEM implemented in the line-by-line radiative transfer model Atmosphit (Coheur et al., 2005) has been used as in Razavi et al. (2011). In Razavi et al. (2011), the reason invoked for performing only forward simulations for HCOOH was the unstable character of the retrievals. With the retrieval settings chosen here, we relied on the retrieved columns since 82% of the selected spectra were successfully retrieved (Tab. 1) and the mean Root Mean Square (RMS) between the observed and fitted spectra is about $2.5\times10^{-6}$ W cm$^{-2}$ sr$^{-1}$ / cm$^{-1}$. This RMS value was close to the IASI estimated radiometric noise.

The conversion factors allowing the calculation of total columns based on $\Delta T_b$ values were determined by using the parameters from a linear regression, obtained by correlating $\Delta T_b$ with HCOOH total columns based on the OEM retrieval. The spectroscopic parameters were from the HITRAN 2008 database (Rothman et al., 2009), using the set of HCOOH spectroscopic line parameters of Vander Auwera et al. (2007). The retrievals have been done with the same a priori profile as in Razavi et al. (2011) but with a loose constraint (350%). The reference channels used for the calculation of $\Delta T_b$ were chosen on both sides of the HCOOH channel (1105 cm$^{-1}$), i.e. at 1103.0 and 1109.0 cm$^{-1}$.

In the OEM-based retrieval, only cloud free scenes (when the cloud coverage for the pixel is below 2%) have been used. The total columns of interfering species in the studied spectral range, such as ozone, ammonia and CFC-12, in addition to the partial columns of water vapor, were retrieved simultaneously. The details of the retrieval parameters are given in Table 2. The EUMETSAT L2 operational data were used, and day-time and night-time data with a positive thermal contrast (TC) were taken into account. The TC was defined as the temperature difference between the surface and the air just above. Negative TC data were excluded, as these were found to deteriorate the correlation $\Delta T_b$-total column.

The OEM-based retrieval has been performed over seven geographical regions shown as blue boxes in Fig. 1. These include five source areas (in Southern Africa, Northern Africa, Amazonia, Australia, Russia) and two remote areas (over the Atlantic and Pacific Oceans). These seven regions are representative of different conditions: emission sources, remote areas, areas influenced by long-range transport, over land and over sea. We have retrieved the five first days of each month in 2009 over the seven regions, allowing the characterization of the seasonal variation. The localization of each area and the number of retrieved spectra are given in Table 1.

From these retrievals, we derived a linear regression between the retrieved total columns and the $\Delta T_b$, illustrated by Fig. 3a. A good correlation was found (r = 0.74) between both parameters. The density of the distribution is also given (Fig. 3b), showing a larger density of data between 0.3 and 0.6 K on the $\Delta T_b$ axis, and between 0.4 and $0.6 \times 10^{16}$ molec/cm$^2$ for the total column. As in Razavi et al. (2011), this relationship was found to depend on the local TC conditions. This dependence is characterized by the color code showing the TC values in Fig. 3a. For example, a $\Delta T_b$ equal to 0.5 K generally corresponds to a total column close to 0 at high TC, or around $2 \times 10^{16}$ molec/cm$^2$ at very low TC.

**2.3 Development of an updated IASI HCOOH dataset**

**2.3.1 Reduction of the thermal contrast dependence**

To account for the TC dependence in the $\Delta T_b$ -total column relationship as shown in Fig. 3a, we have performed forward simulations with different thermal contrasts, using the observation made in the regions and periods listed in Table 1. We artificially modified the surface temperature for each atmospheric profile by $\pm 5$ K. In total, three TC conditions were simulated: $TC_{ref}$, $TC_{ref}+5K$, $TC_{ref}-5K$. These simulations were performed with the a priori as initial total column. In total, 13155 spectra were simulated. The linear regression between TC and $\Delta T_b$ is illustrated in Fig 4a. A fair correlation was found, with r equal to 0.6. Based on this linear regression, a corrected $\Delta T_b$ ($\Delta T_{bTC}$) was defined using equation (1):

$$\Delta T_{bTC} = \Delta T_b - (TC \times a_1 + a_2) \tag{1}$$

with $a_1 = 0.0138$ and $a_2 = 0.3502$. These $\Delta T_{bTC}$ still presented a high correlation with the OEM-based total columns (r = 0.75) and the TC dependence disappeared, as shown in Fig 4b. We then deduced a relationship between the total column and the measured $\Delta T_b$ as below:

$$x = b_1 \times \Delta T_{bTC} + b_2 \tag{2}$$

with $x$ being the total column in $10^{16}$ molec/cm$^2$, $b_1 = 1.5713$ and $b_2 = 0.6792$.

Note that this conversion could lead to negative total columns. If we eliminated all the negative values and kept only all the positive values, we would introduce an artificial bias in the average. For comparisons with zonal or temporal averages, the negative total columns were included in the average. But when the average was found to be negative, it was filtered out.

**2.3.2 Error estimation**

This technique has a low computational cost but the drawbacks of the method are the difficulty to characterize the retrieval in terms of vertical sensitivity (averaging kernels not available) and the lack of an error budget.

The total error of the $\Delta T_b$ approach can be described by three terms: 1) the instrumental error, 2) the error caused by the conversion from $\Delta T_b$ to total column, and 3) the error originating in the OEM-based retrieval. To provide an estimate of the algorithm error, simulations were performed for the data set over the seven regions, using six initial total columns, i.e. by

perturbing the a priori (-50%, the reference a priori, +50%, +100%, +200%, +350%). In total, 26310 forward simulations were used. In the simulations, the temperature profile used was from EUMETSAT operational level 2.

A Gaussian distributed random noise (with $\sigma = 0.15$ K, corresponding to the noise in the studied spectral range) was added to the $T_b$ channels used for the $\Delta T_b$ calculation from the simulated spectra. Then the conversion formula (Eq. 2) was applied to the calculated $\Delta T_b$.

Figure 5 shows the histogram of the relative difference (RD) between the calculated total columns and the true total columns used as input in the forward simulations. The RD was defined as the difference between the calculated total columns and the true total columns, divided by the latter. Positive RDs imply that the calculated total column is higher than the true column. This histogram presents a mean of ~1.6% and a standard deviation around 69%. Note that these results agree with those from Razavi et al. (2011), who found a mean RD equal to -0.8% and a standard deviation of 60%. The relative difference was not impacted by the TC or the $H_2O$ profile but it depended on the HCOOH total column. Fig. 6a shows the dependence of the mean RD on the HCOOH total column used as input to the forward simulation. Large positive RDs (up to 78%) were found for low total columns whereas negative RDs prevailed for large HCOOH columns (lower than 35%). In other words, the retrieval based on brightness temperature differences tends to overestimate the low values of the true columns, and to underestimate the high values.

Considering the detection threshold defined as 2σ on the $\Delta T_b$ (=0.30K), an indicative total column detection threshold was calculated using our conversion factors. To do so, forward simulations were performed for different total columns and TC. The result is illustrated in Fig. 6b and this shows that for the less favorable TC condition (TC=0K) the detection limit of HCOOH is close to $0.6 \times 10^{16}$ molec/cm$^2$ ($0.4 \times 10^{16}$ molec/cm$^2$ for σ). This detection limit is improved with higher TC.

## 3. Analysis of the dataset

### 3.1 Global Distributions

Mean HCOOH global distributions (averaged on a 0.5°×0.5° grid) from IASI for the 2008-2014 period are presented in Fig. 7 and compared with columns obtained using the retrieval method of Razavi et al. (2011). Note that Razavi et al. (2011) retrieved only total columns over land. Except over Indonesia, lower values are observed over the source regions with the updated dataset. The previous section shows that large positive RDs are expected for very low true columns. Even if the columns from Razavi et al. (2011) are not the true columns, this could explain why the total columns for this study are higher over remote areas (e.g. deserts) than those obtained using the methodology described by Razavi et al. (2011). It is also important to note that in Razavi et al. (2011), only averaged data in a 0.5°×0.5° grid with TC higher than 5K were considered. This implies that only data with a strong signal were used, probably overestimating the threshold of the $\Delta T_b$ and thus also the retrieved columns.

Yearly global distributions between 2008 and 2014 with the updated dataset are also presented in Fig. 8 (on a 1°×1° grid).

These distributions highlight well the recurring source regions detected by IASI such as Equatorial Africa, the North of Australia, Amazonia and India, and also the long-range transport such as over the Atlantic Ocean from Africa. The long-range transport over oceans (Atlantic, Indian and Pacific) was not investigated in Razavi et al. (2011). The retrieved columns over the Atlantic Ocean are consistent with the FTIR data from ship cruises reported in the study of Paulot et al. (2011). They showed a gradient of columns from the Poles to the Equator, with the highest values between 0 and 10°N, but with large variability, as the maximum was $3.5 \times 10^{16}$ molec/cm$^2$, but the monthly mean in this region was only $0.5 \times 10^{16}$ molec/cm$^2$.

Several hotspots and distributions are detected and are numbered from (1) to (10) in Fig. 8.

A particularly striking feature is the large hotspot over Russia (close to Moscow) in 2010 as documented by R'Honi et al. (2013), due to intense forest fires during the summer and also in 2012 over Siberia (see label (1)). The current dataset presents a mean total column twice lower ($2.0 \times 10^{16}$ molec/cm$^2$) than the mean derived using the conversion from Razavi et al. (2011) ($4.2 \times 10^{16}$ molec/cm$^2$), within the emission area (50-55°N, 30°-70°E), on 27 July - 27 August 2010, in agreement with the conclusions from R'Honi et al. (2013). Over Russia, other large columns were also found over Sakha Republic and over Khabarovsk Krai, in 2008 and 2012 (see label (2)). It is also worth noting that the North American boreal emissions around the Hudson Bay were larger between 2008 and 2010 compared to other years (see label (3)). Over North America, and especially the US, we observe lower columns over Louisiana and Texas in 2008 and in 2014 compared to the other years (see label (4)); while larger total columns were measured over Northern Australia between 2012 and 2014, in comparison to the period from 2008 to 2010 (see label (5)).

The monthly means over the seven years are also presented with an animation (Fig. S1) in the Supplement. As already observed over Eastern Russia with label (2), in June 2010 and 2012, there were large concentrations, close to Khabarovsk Krai, compared to the other years in this region. Whereas intense fires were detected in June 2012 in this region, this was not the case in June 2010 (see maps on http://lance-modis.eosdis.nasa.gov/cgi-bin/imagery/firemaps.cgi). The absence of forest fires and the lack of hotpots in the biogenic emission inventory (Fig. 1) in this region points to the presence of an unidentified source, possibly of biogenic origin.

Large columns were similarly retrieved over a large region encompassing Laos, Thailand and Myanmar in April 2010, 2012, 2013 and 2014 (see label (6)). It matches well with the locations of fire hotspots detected by MODIS.

Over India, the largest total columns are observed from March to June probably due to biomass burning (see label (7)). Indeed, those emissions present a marked seasonal variation with a maximum in March-May according to the GFED3 inventory (van der Werf et al., 2010), with 50%, 22% and 11% of annual emissions occurring in March, April and May, respectively.

Larger total columns were retrieved in August 2010 along the Euphrates River compared to the other years (see label (8)).

The monthly distributions also highlight hotspots over the US, besides those shown in the annual distributions (see label (9)). In summer 2011, large signatures over the US were not confined to coastal regions; high total columns were also detected in the Mid-Western US such as over Kansas, Mississippi, Missouri or Oklahoma. These states are flagged as biogenic emission regions of VOCs by Millet et al. (2015), acting as secondary source of HCOOH. In July 2012, the emissions over the US were mostly confined to the Eastern part.

The Asian HCOOH outflow is well captured over the western Pacific (see label (10)). The range of values of the IASI total columns, from 2008 to 2014, broadly agrees with our estimation of total columns using the measurements from the aircraft PEM-West-B campaign conducted in February-March 1994 (Talbot et al., 1997a; 1997b) over a large region covering the latitudes 0°-60°N and the longitudes 110°-180°E. Indeed, measured HCOOH mixing ratio profiles during the campaign mostly ranged around 100-150 pptv from the boundary layer to about 12 km altitude, with peak values of up to 4 ppbv in fresh (< 2 days) plumes originating in China. Using these profiles, we estimated that this corresponded approximately to columns ranging from 0.2 to $0.9 \times 10^{16}$ molec/cm$^2$ while the IASI mean column is around $0.55 \times 10^{16}$ molec/cm$^2$. Over the remote Pacific, the IASI total columns, for the studied period, are larger than measured during the aircraft PEM-Tropics-A campaign in August-December 1996 (e.g. Talbot et al., 1999). They measured mixing ratios of the order of 20-40 ppbv in the boundary layer and 50-100 pptv in the free troposphere, corresponding to estimated total columns of $0.1-0.2 \times 10^{16}$ molec/cm$^2$. This overestimation is in agreement with the error budget from Fig. 6.

Overall, these monthly means highlight the seasonal variation of the HCOOH distribution around the world. The animation reveals clear variations in the HCOOH distribution due to the seasonality of biomass burning and vegetation growth. It is well shown with the large total columns observed during September and October 2008, 2012, 2014 in the Southern Hemisphere (over Amazonia, Africa and Australia). In 2010, the same features were noted except for Australia.

### 3.2 Comparison with ground-based FTIR measurements

The IASI HCOOH retrieved columns in this work have been compared with ground-based FTIR measurements. This comparison was done without smoothing the data since the averaging kernels (AKs) were not provided by our retrieval method. This comparison is presented at four sites: Jungfraujoch (46.55°N 7.98°E) in Switzerland, Wollongong in Australia (34.41°S 150.88°E), Saint-Denis (20.88°S 55.45°E) and Maido (21.07°S 55.39°E) at La Réunion (Fig. 9). The current retrieved columns have also been evaluated with those using the methodology from Razavi et al (2011) over the same sites (Fig. 10).

A complete description of the FTIR instruments and the retrieved HCOOH data can be found in Zander et al. (2010), Paton-Walsh et al. (2005), and Vigouroux et al. (2012), for the Jungfraujoch, Wollongong and Saint-Denis stations, respectively. For the Jungfraujoch, spectra were typically recorded at spectral resolutions of 0.004 and 0.006 cm$^{-1}$. For the present subset, a mean signal-to-noise ratio (SNR) of 895 was computed, with 10-th and 90-th percentiles of 525 and 1525, respectively. A

uniform scaling of the HCOOH a priori was performed, and no information was available regarding the sensitivity of the retrieval with altitude. For Wollongong, the spectral resolution was 0.004 cm$^{-1}$ and the SNR was around 1000-2000. Over La Réunion, the HCOOH retrieval parameters were the same for Saint-Denis and for the more recent data at the Maïdo station. The spectral resolution of La Réunion spectra was 0.007 or 0.011 cm$^{-1}$, depending on the time of the measurement. The SNR was about 1000-2000 depending on the spectra.

The current IASI retrieved columns were also compared with a set of columns retrieved by OEM around the sites. For each OEM-based retrieved column, the corresponding column using the conversion factors was calculated, showing that the current dataset and the OEM-based retrieval are in agreement (correlation ranging from 0.7 to 0.8, with an underestimation of the columns calculated with the conversion factors between 6 and 15%) (Fig. S2). It is also worth noting that similar biases were found between the columns retrieved by OEM around the ground-based locations and the FTIR columns as between the columns retrieved in this work and the FTIR ones (Tab. S1).

Averaging kernels were also unavailable for the FTIR measurements performed at Jungfraujoch (extended from Zander et al., 2010) and at Wollongong (Paton-Walsh et al., 2005). The measurements at Saint-Denis and Maido reached a maximum sensitivity between about 3 and 12 km as described in Vigouroux et al. (2012) and shown in Fig. 11. The AKs indicate the vertical sensitivity of the retrieval. The Jacobians express the sensitivity of the radiative transfer model and the IASI instrument (through its instrumental function) to the variation of HCOOH in the atmosphere. Both functions then give a good indication of the vertical sensitivity for each data set. These AKs and the Jacobians show that FTIR and IASI were both mostly sensitive to the free troposphere but that the FTIR measurements presented a broader vertical sensitivity, reaching higher altitudes than IASI.

A difficulty in comparisons of satellite columns with ground-based measurements over mountain sites like Jungfraujoch over the Swiss Alps (3.6 km altitude) or Maido at La Réunion (2.2 km altitude), is the difference of altitude between the FTIR sites and the co-located IASI ground pixel height. To account for the altitude dependence, both the IASI and the FTIR total columns were normalized to the sea level altitude using:

$$C_{corrected} = C \times \exp(H/7.4) \tag{3}$$

where H represents the ground measurement height (in km) and C is the total column (in $10^{16}$ molec/cm$^2$). This simplified formula is a variation of the hypsometric equation (Wallace and Hobbs, 1977). This normalization relies on the assumption that the HCOOH mixing ratio is constant as a function of altitude. Although crude, this procedure improved the comparisons.

The time series of the IASI and FTIR columns over the selected sites are shown in Fig. 9. The comparison used IASI data collocated within 0.5° of the site location in both latitude and longitude. To keep enough IASI data to compare, daily averages were used. A more stringent criterion of ±2h was tested but provided similar results, except over Maido where the correlation increased to 0.6 without improvement of the bias. The advantage of this daily average is the possibility to derive

the seasonal variation over each site. Over all sites, the broad patterns of seasonal and inter-annual variations were similarly captured by IASI and the ground-based FTIR.

The comparison between the ground-based measurements and the total column derived from the IASI spectra may be affected by sampling differences associated e.g. with cloudiness. IASI may be able to measure through clear skies in the vicinity of the station when the FTIR data are not available due to localized cloud. Moreover, despite the use of strict co-location criteria (spatial and temporal), most mismatches in peak values could be a result of mismatches in the spatial and temporal scales of the measurements being compared.

The correlation coefficients and the biases between FTIR and IASI are also provided for each year in Table 3. Over Jungfraujoch (Fig. 9), large total columns were measured by the ground-based instrument during the spring and the summer for each year. These large values were not captured by IASI data, causing a large bias of $1.14 \times 10^{16}$ molec/cm$^2$ on average. The comparison at Jungfraujoch presents the lowest correlation coefficient among the four stations. This might be caused to some extent by the error associated with the normalization to the sea level, which is largest at the high altitude of the Jungfraujoch.

The seasonal cycle obtained from IASI agrees well with FTIR data over both sites at La Réunion (correlation coefficient r = 0.77 at St-Denis, up to 0.85 in 2011; see Table 3) but the columns from the IASI retrieval show a large positive bias at both sites (>100%). The overestimation by IASI is especially large for background columns, i.e. between February and July, when FTIR columns are of the order of $0.15 \times 10^{16}$ molec/cm$^2$. This increase of the bias for lower values of the true column is qualitatively consistent with the dependence of the error associated with the conversion from brightness temperatures on the magnitude of the true column and especially for lower TC (Fig. 6): the expected error is lower than 20% for columns ranging between 0.5 and $1 \times 10^{16}$ molec/cm$^2$, and close to 80% for columns lower than $0.3 \times 10^{16}$ molec/cm$^2$. In addition, the difference in the altitude range of the vertical sensitivity between IASI and the FTIR could also contribute to the biases at Saint-Denis and Maido. Regarding the time series at La Réunion, it is worth noting that large columns measured in October 2010 and 2011 over Saint-Denis, corresponding to enriched plumes from Africa, are seen in the spatial distributions (Fig. S1).

At Wollongong, IASI and ground-based FTIR background levels are in broad agreement. The correlation is highest in 2008. The peaks in the HCOOH columns in October 2012, 2013 and in Nov 2014 observed by both instruments are also seen in the distributions in Fig. S1. As in the case of La Réunion data, a larger positive bias is found when the FTIR total columns are low (<$0.5 \times 10^{16}$ molec/cm$^2$).

The FTIR measurements were also used to evaluate the current HCOOH columns with those using the conversion from Razavi et al. (2011) (Fig. 10). The colocation criteria have been enlarged to ±4° as used in the evaluation shown in Stavrakou et al. (2012). The criterion was enlarged since the number of available data from Razavi et al. (2011) around the sites was less important than for the current dataset. Fig. 10 shows the distribution of the relative differences with the FTIR measurements for both methodologies. It provides information about the bias, the normalized bias and the correlation

coefficient. At all sites, the distribution is more spread out with the conversion from Razavi et al. (2011). The correlation coefficient is largely improved with the updated dataset, except over Saint-Denis where it is similar. The bias is also significantly reduced with the updated dataset except over Jungfraujoch. This difference over Jungfraujoch is coherent with the previous comparison (Fig. 9) since the updated dataset is underestimated compared to the FTIR measurements.

Overall, the current dataset presents higher correlation and lower bias than the columns from Razavi et al. (2011).

### 3.3 Comparison with IMAGESv2

The IMAGESv2 global CTM runs at 2° resolution in latitude and 2.5° resolution in longitude. The model is resolved at 40 vertical levels, from the surface up to 44 hPa (Stavrakou et al., 2011). The biogenic emissions of isoprene (believed to be the most important precursor of HCOOH) were obtained from MEGAN-MOHYCAN (Stavrakou et al. 2014). The vegetation fire emissions were from GFEDv3 (van der Werf et al., 2010). This dataset distinguished emissions from savanna, woodland, and forest fires, agricultural waste burning, peatlands, deforestation and degradation fires. Anthropogenic emissions were constructed from a mix of inventories: REASv2 in Asia (Kurokawa et al. 2013), NEI in USA (from www.epa.gov/ttnchie1/trends/), EMEP (obtained from www.ceip.at/webdab-emission-database/emissions -as-used-in-emep-models/) in Europe, and the RETRO database (Schultz et al., 2007) for the rest of world.

### 3.3.1 Seasonal variation

For the sake of comparison, the IASI HCOOH total columns have been averaged to the horizontal model grid resolution. The IASI and the model total columns have also been averaged by season, defined as: December-January-February (DJF), March-April-May (MAM), June-July-August (JJA) and September-October-November (SON) over the seven years, between 2008 and 2014. Figure 12 presents these global distributions for IASI and the model. For each season, we find that the IASI total columns are higher than those from IMAGESv2 simulations. This highlights the difficulty to predict the measured concentrations by models as found in previous modelling studies such as Stavrakou et al. (2012) and Paulot et al. (2011).

During winter (DJF), the model shows large total columns over Equatorial Africa and Asia while IASI only detects large values over Africa.

In spring (MAM), the CTM largely underestimates the distributions over Africa compared to IASI.

In summer (JJA), the emissions from fires seem to be the primary cause for the strong HCOOH enhancements in the CTM (Africa, Amazonia, boreal regions: Canada and Eastern Siberia) although biogenic emissions (Southeastern US) and anthropogenic activities (Eastern China) also produce visible enhancements. In the IASI distributions, enhancements associated with fires are less prominent over North and South America. IASI reveals larger total columns over Africa in JJA than over South America while the CTM shows roughly the same values over both regions. Compared with the CTM, IASI

also shows larger HCOOH columns over the Midwestern US, India and semiarid regions in Southwestern Russia and Kazakhstan.

During fall (SON), the columns from IASI over the southern hemispheric biomass burning emission regions (South America, Africa and Australia) are larger than over Asia (India and China) and over Indonesia while those simulated by the model are quite similar for these regions.

### 3.3.2 Inter-annual variation

The global and seasonal distributions from IASI suggest an underestimation of the modelled columns and a misrepresentation of some emission sources. Figure 13 presents the time series over the areas identified by black boxes in Fig. 1. These regions are: AMER (North America), AMAZ (Amazonia), AFRI (Africa), SIBE (Central Siberia), INDI (India), ASIA (Asia) and AUST (Australia).

Over all regions, these time series confirm a large bias of IASI in comparison to the simulations. The seasonal variation is,
however, well represented, except over India, where the seasonal cycle is out of phase. This phase difference is coherent with the cycle shown on the global distribution (Fig. 12). The model underestimation over India suggests that either the emission factors of HCOOH (or its precursors) are underestimated in the model, or the biomass burnt estimated by GFEDv3 is too low for this region. The underestimation for forest fire emissions is a severe issue as shown by Chaliyakunnel et al. (2016) over the tropical forests. The calculated linear trend is also provided, based on the annual IASI mean (blue dots in
Fig. 13). Over three regions (North America, Africa and Asia), the trend is negligible (0.2-0.4%/year) but a small increase is noted over Australia, Amazonia and India (1.5±0.5%/year, 0.9±0.7%/year and 0.7±0.1%/year respectively). A decrease of 0.8±0.9%/year is also observed over Siberia.

Over Amazonia, the large peak in 2010 due to large forest fire emissions (e.g. Hooghiemstra et al. 2012) is well represented in the model and captured by IASI; however, the peak in the IASI data occurs one month later than in the CTM. A similar
shift was already observed between the CO and $NH_3$ observations from IASI (Withburn et al., 2015). This HCOOH shift could be partly due to the difference of altitude in the HCOOH vertical distributions. IASI detects the plumes in the free troposphere, whereas the IMAGES columns reflect more directly the surface emissions. The mean altitude of the maximum in the HCOOH vertical profile from the model is located around 2.8 km for this region and close to 2 km in August 2010, while the maximum of vertical sensitivity from IASI is located at higher altitudes. This assumes that transport time could be
close to one month but it is longer than the known mean transport time between the boundary layer and the upper troposphere. Over Siberia, the summer 2010 and at a lesser extent the summer 2012, were exceptional in terms of large HCOOH emissions, as evident in the global distributions shown in Fig. 8. As revealed by the correlation coefficients, the seasonal cycle is very well captured over Asia (r=0.75), Australia (r=0.92), Siberia (r=0.94) and North America (r=0.94).

The double peaks (~March and September) over Africa are explained by the shift of emission seasons on either side of the Equator. Over Africa, the correlation is lower, due to two factors: there is a shift of one month between IASI and the CTM in the maximum of columns, and the higher columns from IASI remain longer in time than those simulated (especially between the peak around January with the CTM and the one around February-March for IASI). A difference of altitude in the detection of HCOOH between the CTM and IASI could not explain IASI detected higher total columns during more than one month in the troposphere in March. Around March-April, the mean altitude of maximum in the simulated columns is around 4 km, close to the altitude of maximum vertical sensitivity from IASI, showing that the CTM and IASI should detect the same enriched HCOOH plume.

## 4. Conclusions

Global distributions of HCOOH were derived from IASI radiance spectra, using conversion factors between representative retrieved total columns and selected radiance channels ($\Delta T_b$). This paper presents seven years of HCOOH measurements recorded by IASI. A limitation of this dataset is its lack of characterization for vertical sensitivity due to the conversion technique used, even though the maximum sensitivity is shown to be in the mid-troposphere. This approach has the significant advantage of reducing the computing time necessary to analyze large amounts of data and to provide a global representation of the HCOOH distribution around the world, including ocean and land scenes, and emission sources as well as remote areas.

IASI provides global distributions of HCOOH, highlighting the long-range transport of tropospheric HCOOH over the Atlantic Ocean and the detection of source regions, e.g. biomass burning areas over Amazonia, Africa, Australia and Siberia. Other source regions are detected such as the Mid-eastern United States in 2011 or over India.

The comparison with an atmospheric model and, to a lesser extent, with ground based FTIR observations remains challenging. Despite large biases in many cases, we show that the inter-annual and the seasonal variations are well captured by IASI when compared with ground-based FTIR measurements and the IMAGESv2 CTM. The best overall correlation with the FTIR is obtained at Saint-Denis over La Réunion (r=0.77) but both sites at La Réunion present the larger biases (>100% lower than IASI). High correlations are obtained with the CTM, in particularly over America, Australia, Siberia and Asia, with correlation coefficients ranging from 0.75 to 0.94. This comparison also points out a misrepresentation of the distribution between IASI and the CTM over India and Africa and a global underestimation of the distribution by the CTM. A small decreasing trend during the 7-yr period is observed over Siberia, while a small increase is noted over India, Amazonia and Australia (0.7%/year, 0.9%/year and 1.5%/year respectively).

The HCOOH columns from IASI will require further evaluation and probably improvements to narrow down the biases but the dataset available spans now seven years and it will likely contribute to a better understanding of the tropospheric HCOOH budget. The dataset will be made available through the Ether database (http://ether.ipsl.jussieu.fr) for further

scientific studies. This 7-yr record will be completed by the data provided by IASI/MetOp-B, launched at the end of 2012, and IASI/MetOp-C to be launched in 2018. The IASI program will be followed up (after 2022) by the IASI-NG mission aboard the MetOp-SG satellite series (Clerbaux and Crevoisier, 2013; Crevoisier et al., 2014). This new instrument will be characterized by improved spectral resolution and lower radiometric noise. It will lead to a better vertical resolution, along with improved accuracy and detection threshold.

**Acknowledgments.** IASI was developed and built under the responsibility of Centre National des Etudes Spatiales (CNES) and flies onboard the MetOp satellite as part of the EUMETSAT Polar system. We thank the Aeris data infrastructure (http://www.aeris-data.fr/) for providing access to the IASI L1C data and to the ECCAD emission inventories used in this study. The French scientists are grateful to CNES for financial support. We also gratefully acknowledge Yasmina R'Honi and Lieven Clarisse (ULB) for their help with the HCOOH retrieval. The authors would like to thank the other researchers involved in the ground-based solar remote sensing program at Wollongong (including Nicholas B. Jones, David Griffith, Nicholas Deutscher and Voltaire Velazco) and the Australian Research Council for its support for the NDACC site at Wollongong, most recently as part of project DP110101948. Acknowledgements are addressed to the Université de La Réunion and CNRS (LACy-UMR8105 and UMS3365) for their support of the OPAR station (Observatoire de Physique de l'Atmosphère de la Réunion). The authors gratefully acknowledge C. Hermans, F. Scolas, and B. Langerock from BIRA-IASB, and J.-M. Metzger from Université de La Réunion, for the Reunion Island measurements. The University of Liège contribution to the present work has been supported by the F.R.S. – FNRS, the Belgian Science Policy Office (BELSPO), Brussels, the Fédération Wallonie-Bruxelles and MeteoSwiss (GAW-CH program). We thank the International Foundation High Altitude Research Stations Jungfraujoch and Gornergrat (HFSJG, Bern). We are grateful to all Belgian colleagues who contributed to the acquisition of the FTIR data at Jungfraujoch. The Reunion Island measurements have mainly been supported by the A3C project (PRODEX Program of BELSPO). The research was supported by BELSPO and ESA through the IASI.Flow project (Prodex arrangement 4000111403).

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

**Table 1** Selected regions used for the retrieval. The localization of each area and the number of spectra retrieved during the studied period are provided. The numbers correspond to the total number of successfully retrieved spectra and those given in parentheses to the total number of spectra in each region.

| Region | Localization | Number of retrieved spectra |
|---|---|---|
| Africa N | 6-7°N 18-22°E | 265 (358) |
| Africa S | 12-14°S 20-24°E | 788 (1083) |
| Amazonia | 6-10°S 43-45°W | 682 (739) |
| Atlantic | 22-24°N 42-45°W | 675 (737) |
| Australia | 14-15°S 131-133°E | 218 (271) |
| Pacific | 20-22°S 140-142°E | 472 (492) |
| Russia | 50-54°N 60-62°E | 538 (781) |


**Table2** Retrieval parameters used in this work.

| Parameters | Layers | A priori | Variability |
|---|---|---|---|
| $H_2O$ | 0-10 km: five 2 km thick layers 10-20 km: two 5 km thick layers | EUMETSAT operational level 2 | 20% |
| $O_3$ | total column | standard atmosphere | 20% |
| $NH_3$ | ditto | standard atmosphere | 20% |
| HCOOH | ditto | Razavi et al. (2011) | 350% |
| CFC-12 | ditto | Coheur et al. (2003) | 20% |

**Table 3** Correlation coefficient (in italics), mean bias in $10^{16}$ molec/cm$^2$, normalized mean bias (square brackets) in percent, between the daily FTIR measurements and the IASI co-located data. The IASI data were chosen at ±0.5° around the site location. The number of coincidence days is given in parentheses.

| FTIR station | 2008 | 2009 | 2010 | 2011 | 2012 | 2013 | 2014 |
|---|---|---|---|---|---|---|---|
| Jungfraujoch | *0.33* -1.10 [-71.8] (57) | *0.06* -1.21 [-72.9] (53) | *0.23* -1.26 [-73.7] (69) | *0.49* -0.99 [-69.0](91) | *0.47* -1.15 [-72.4] (97) | *0.54* -1.24 [-71.4] (106) | *0.43* - 1.06 [-71.1] (78) |
| Wollongong | *0.77* -0.07 [-11.2] (50) | *0.16* 0.01 [2.9] (79) | *0.56* 0.03 [6.1] (44) | *0.60* -0.02 [-3.5] (96) | *0.63* -0.01 [-1.1] (124) | *0.69* -0.03 [-5.1] (106) | *0.57* 0 [0.6] (56) |
| Saint-Denis (La Réunion) | - | *0.69* 0.28 [110.1] (82) | *0.75* 0.24 [100.1] (72) | *0.85* 0.27 [100.6] (97) | - | - | - |
| Maïdo (La Réunion) | - | - | - | - | - | *0.35* 0.28 [131.2] (60) | *0.53* 0.28 [117.7] (49) |


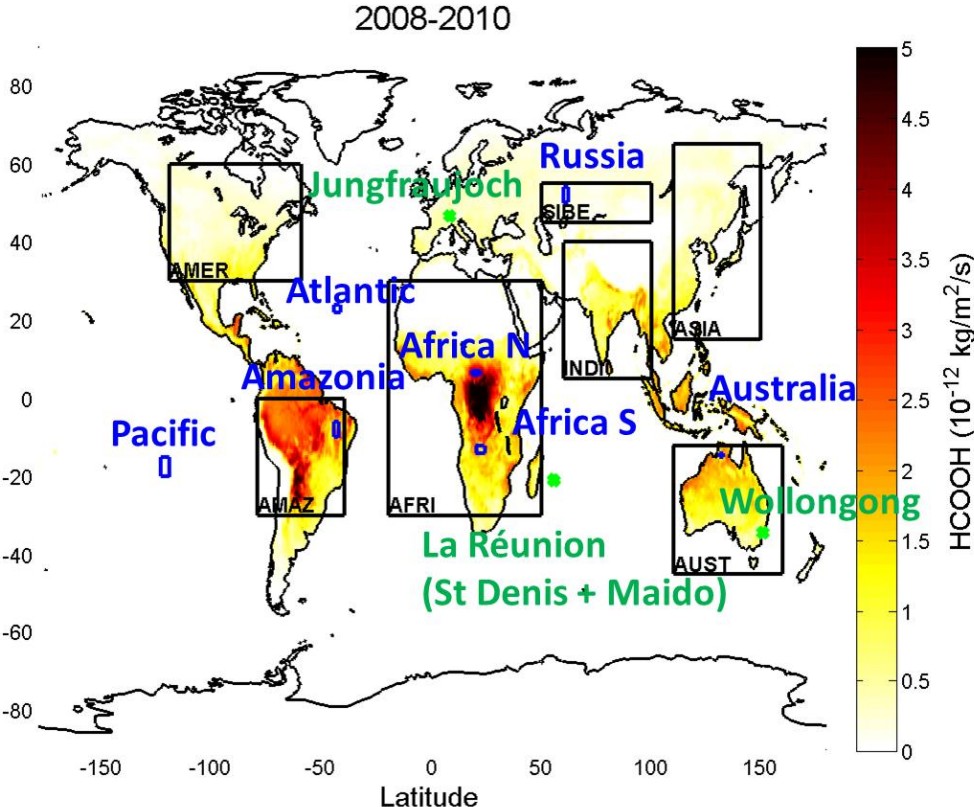

**Figure 1.** MEGAN-MACC HCOOH emissions for the period between 2008 and 2010 on a 0.5°×0.5° grid. The green stars correspond to the location of the FTIR measurements, the 7 selected regions used for the retrievals and described in Table 1 are highlighted in blue, and the black boxes are the regions used for the comparison with IMAGESv2.


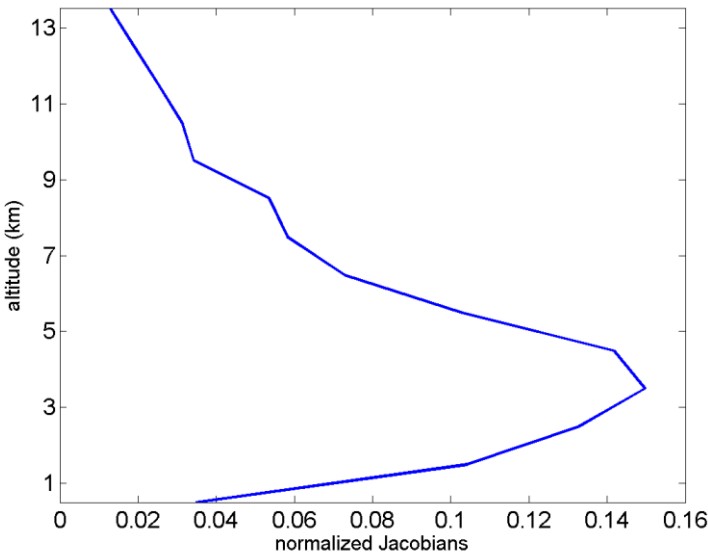

**Figure 2.** Mean normalized Jacobians of all retrieved spectra (over the 7 selected regions) as a function of altitude.


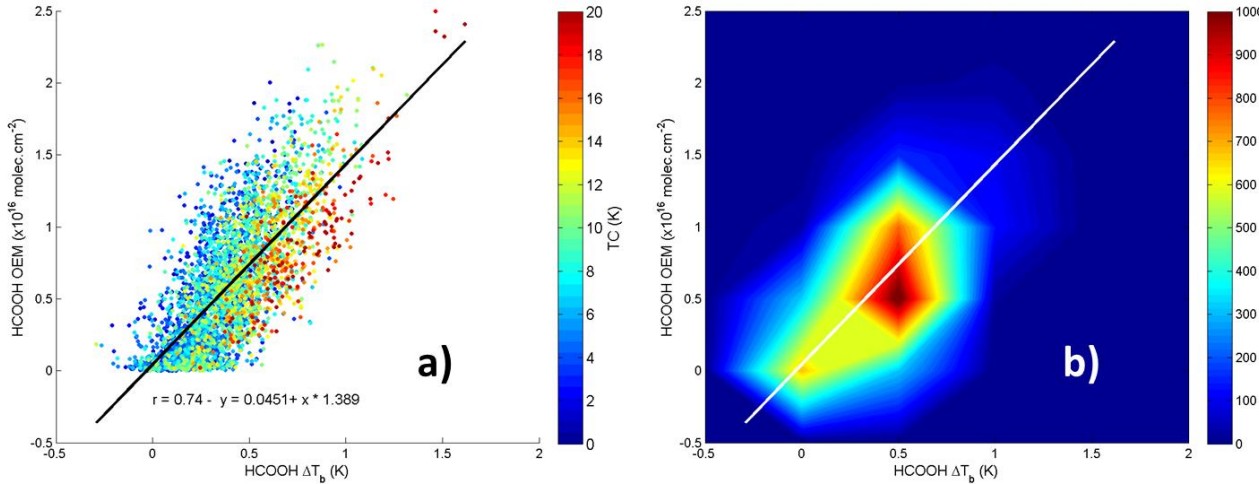

**Figure 3.** a: Scatter plot between all the retrieved total columns using the Optimal Estimation Method (OEM) and the corresponding HCOOH $\Delta T_b$. The black line corresponds to the linear regression. Each data point is colored based on its thermal contrast value. b: Distribution of points density from the scatter plot. The number of points is highlighted with the color scale. The linear regression is also reported by a white line.


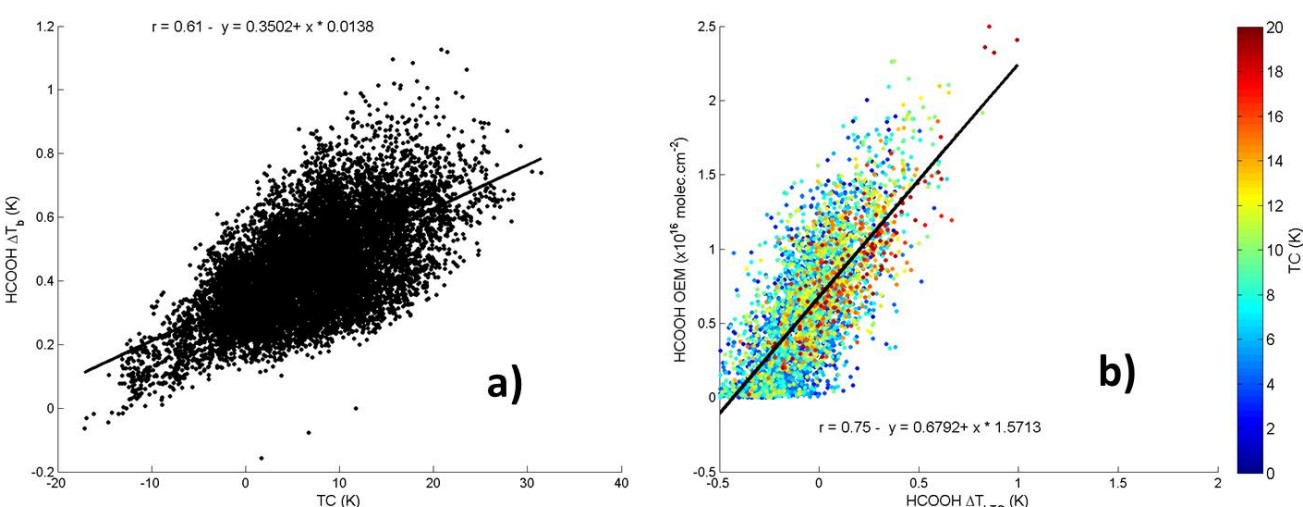

**Figure 4.** a: Scatter plot between the simulated $\Delta T_b$ and the TC for one fixed HCOOH total column ($0.6 \times 10^{16}$ molec/cm$^2$). b: Scatter plot between the HCOOH retrieved total columns using the Optimal Estimation Method (OEM) and the corrected HCOOH $\Delta T_{bTC}$. $\Delta T_{bTC}$ corresponds to the HCOOH $\Delta T_b$ taking account the TC dependence, given by the equation on panel (a). Each data point is colored based on its thermal contrast value as in Fig. 3a.


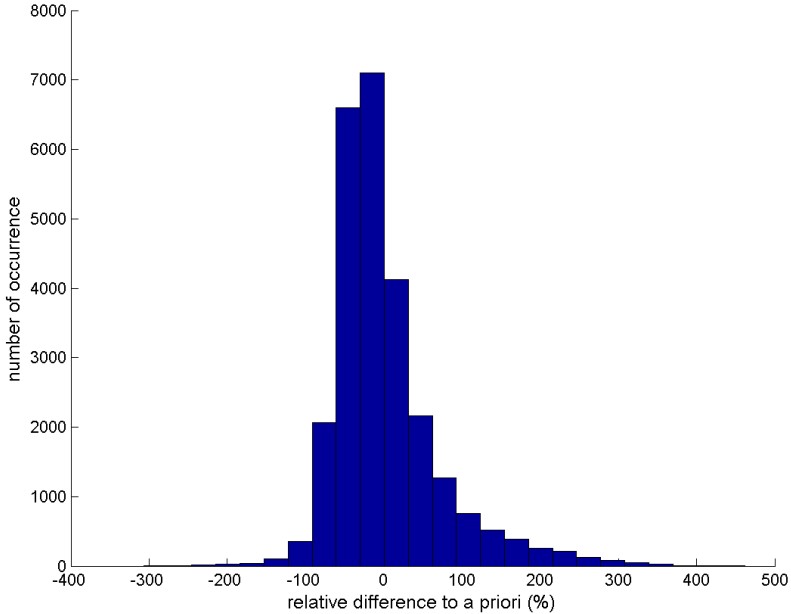

**Figure 5.** Histogram of the relative differences between the calculated total columns (derived from the $\Delta T_b$ conversion) and the a priori total columns (used as input in the forward simulations). The a priori total columns are defined as the reference.

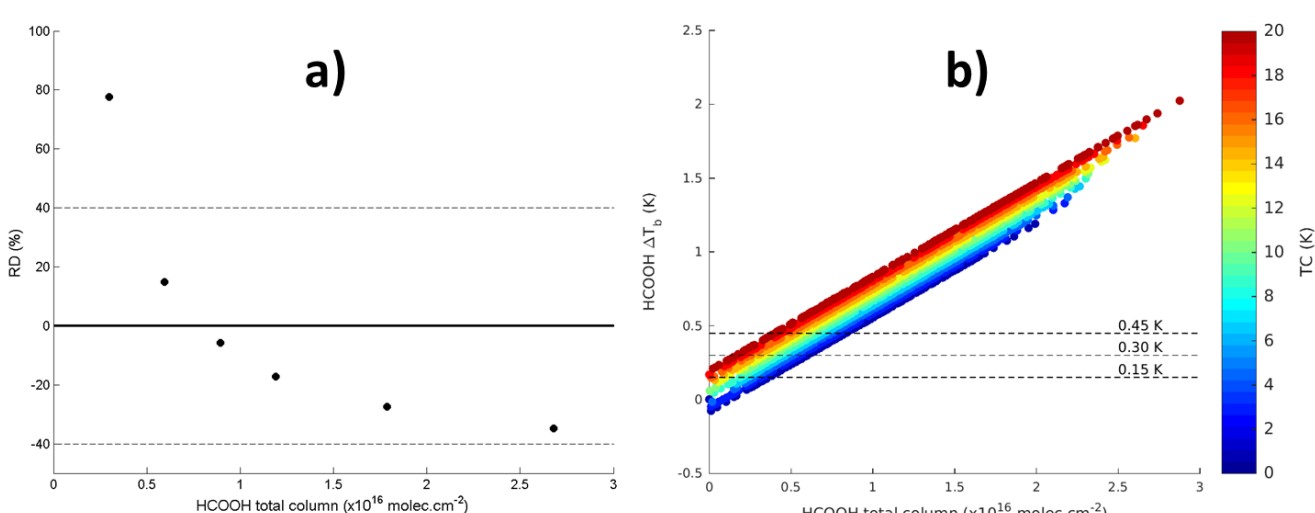

**Figure 6.** a: Variation of the mean relative difference between the total columns derived from the $\Delta T_b$ conversion and the a priori total columns (used as input in the forward simulations) according to the a priori used. The black solid line corresponds to a relative difference equal to 0 and the dashed black lines to ±40%. b: Variation of the simulated $\Delta T_b$ for different HCOOH total columns and TC. The dashed black lines correspond to a $\Delta T_b$ equal to 0.15K, 0.30K and 0.45K. 0.15 K corresponds to the IASI radiometric noise in the HCOOH spectral range (see Section 2.1).

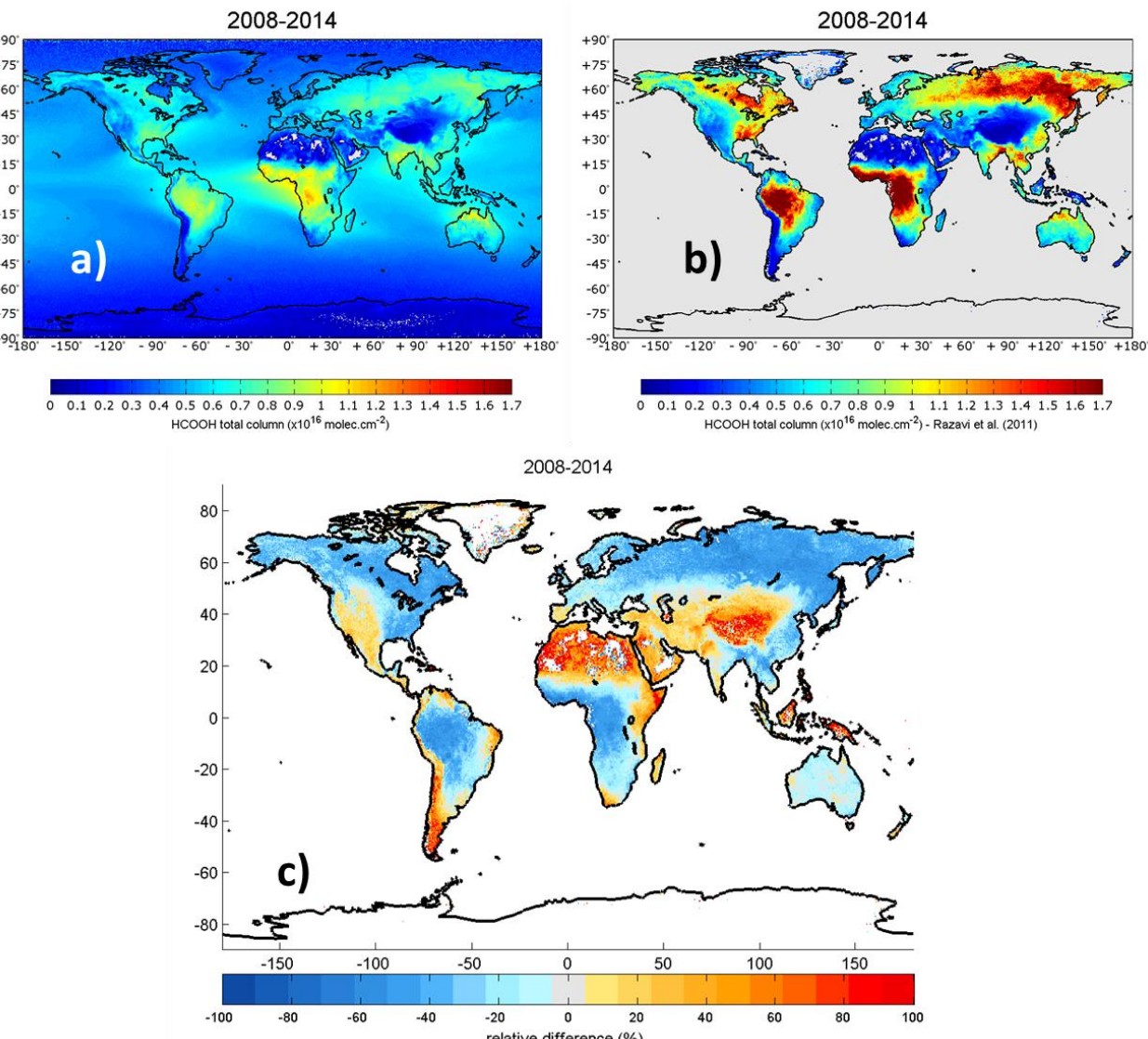

**Figure 7.** Mean HCOOH global distribution between 2008 and 2014, derived using the IASI radiance observations on a $0.5°×0.5°$ grid with the retrieval from this work (a), using the methodology described by Razavi et al. (2011) (b) and the relative difference between both distributions in percent (c). The relative difference is defined as: (HCOOH $_{\text{this work}}$ − HCOOH $_{\text{Razavi et al. (2011)}}$) / HCOOH $_{\text{Razavi et al. (2011)}}$.


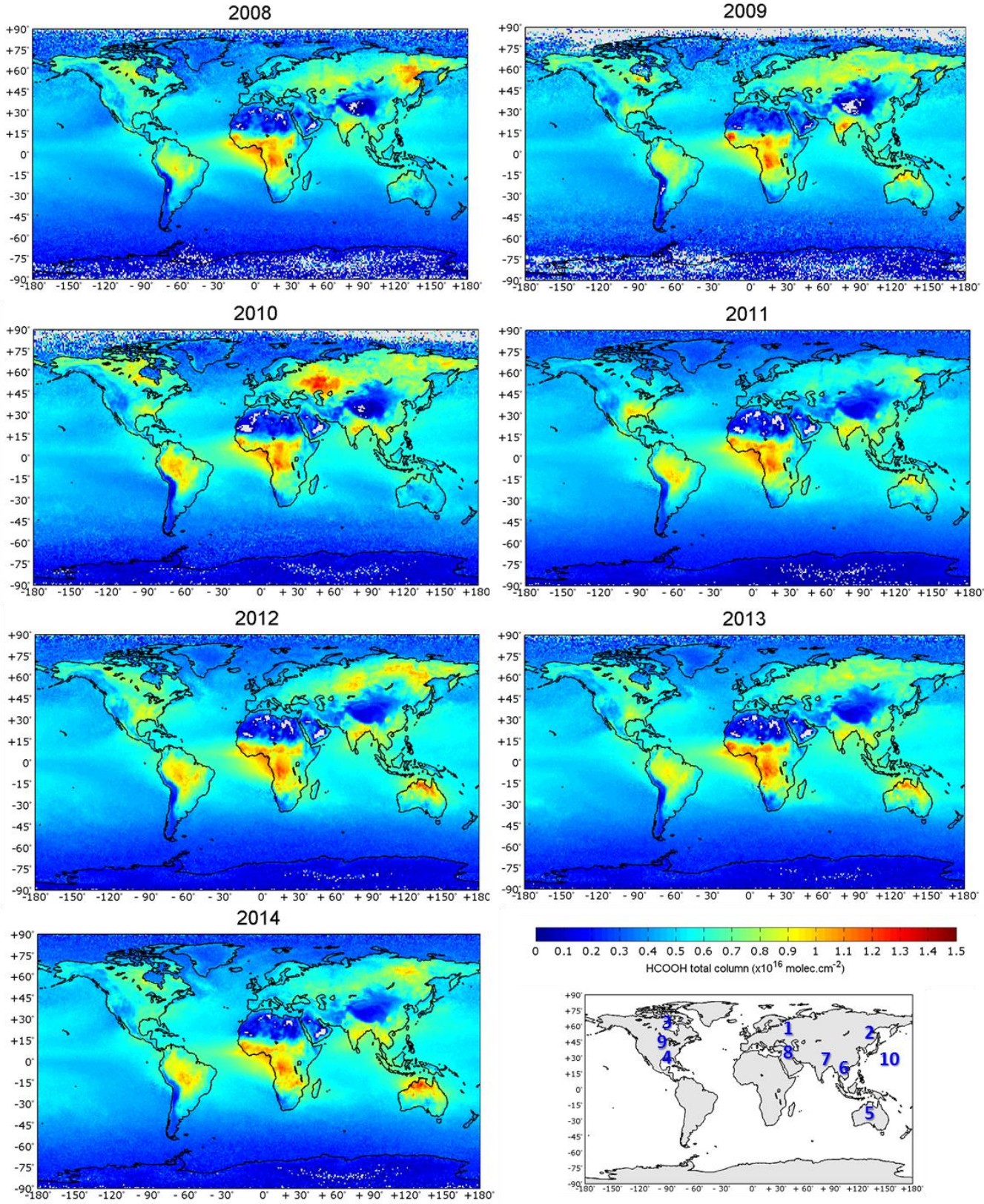

**Figure 8.** Annual HCOOH global distribution from 2008 to 2014, derived using the IASI radiance observations on a 1°×1° grid. Different sources or distributions described in the text are numbered in blue on the bottom-right map.


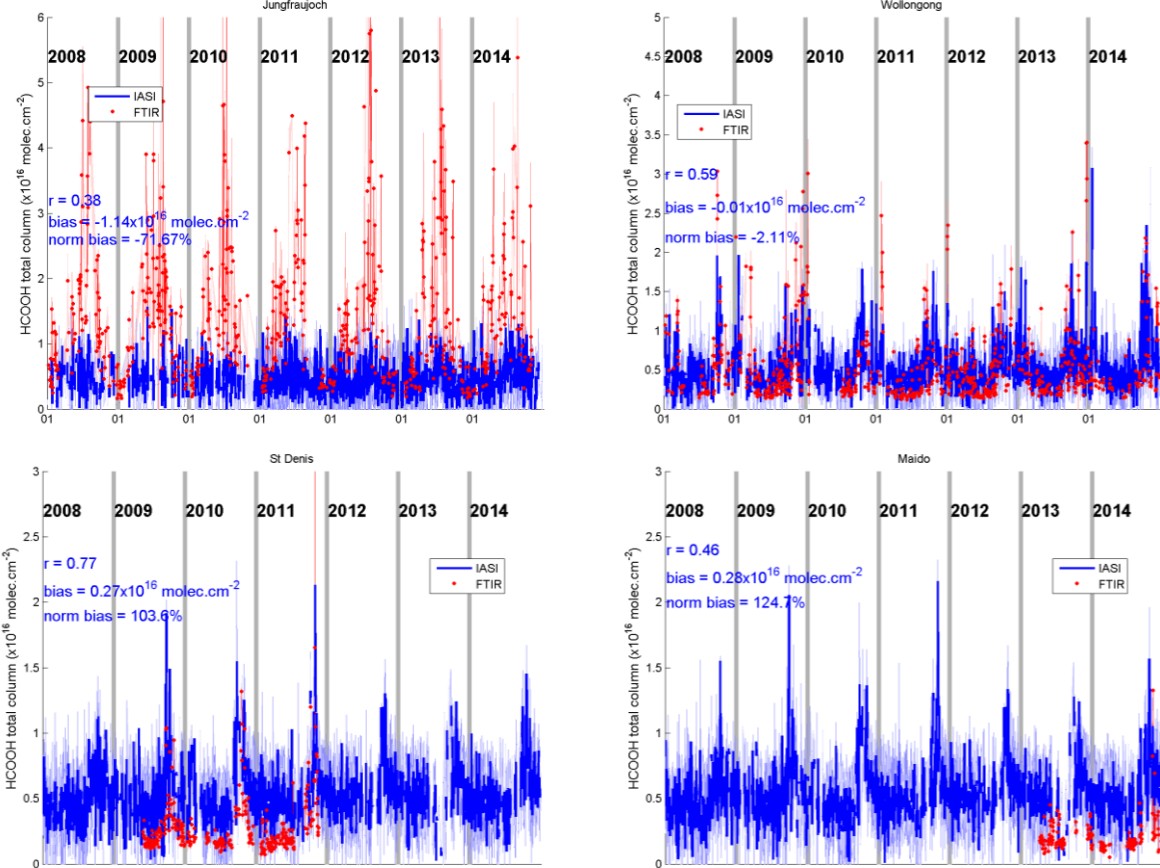

**Figure 9.** Time series of HCOOH daily means over Jungfraujoch (top-left), Wollongong (top-right), Saint-Denis (bottom-left) and Maido (bottom-right) between 2008 and 2014 for IASI (blue) and the ground-based FTIR (red) measurements. The IASI data are collocated at ±0.5° around the site location. The correlation coefficient, the mean bias and the normalized mean bias for all years are given in blue font on each plot. The blue shade error bar corresponds to the standard deviation on the IASI daily means. The altitude of the stations is: 3.6 km for Jungfraujoch, 20 m for Wollongong, 50 m for Saint-Denis, 2.2 km for Maido.

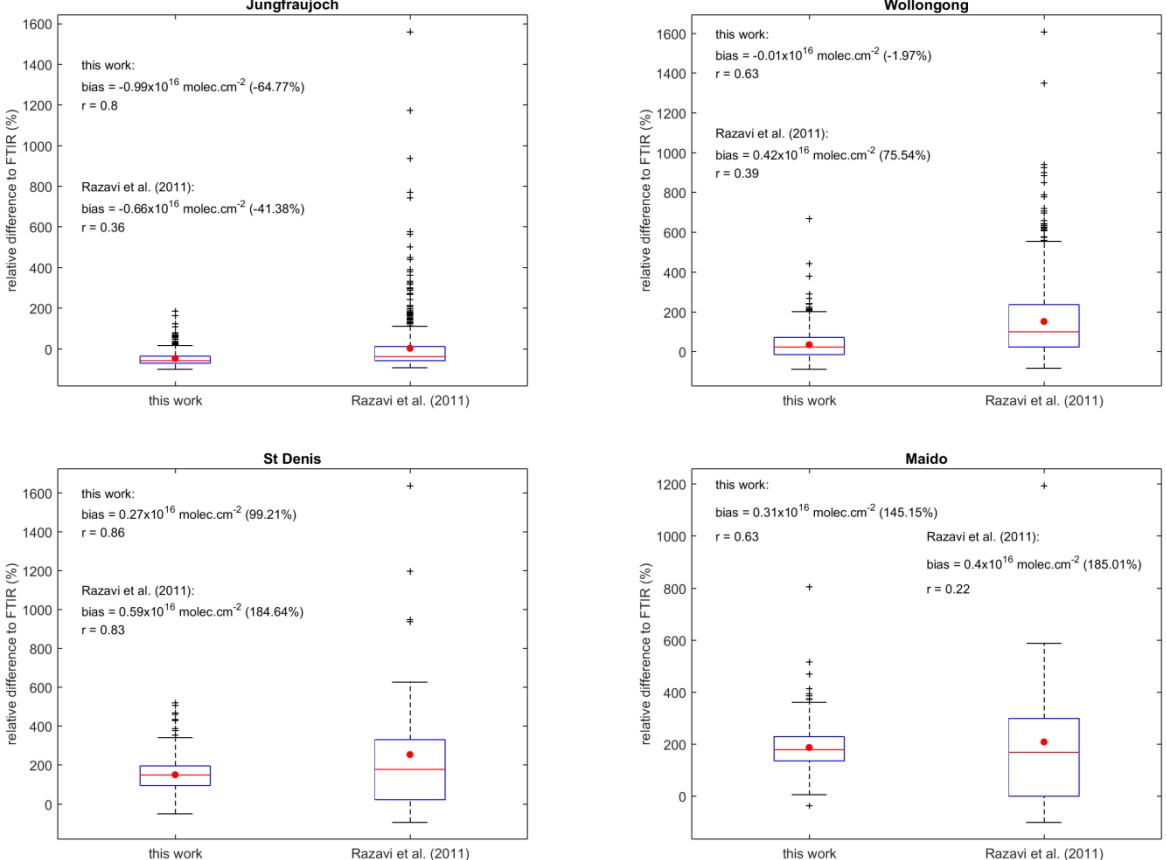

**Figure 10.** Box and whisker plots showing mean (red central circle), median (red central line), 25[th] and 75[th] percentile (blue box edges) of the relative difference between the HCOOH derived using the IASI radiance observations from this work or using the conversion from Razavi et al. (2011) and the FTIR measurements for each station: Jungfraujoch, Wollongong, Saint-Denis and Maido. The whiskers encompass values from 25[th]−1.5×(75[th]−25[th]) to the 75[th]+1.5×(75[th]−25[th]). This range covers more than 99 % of a normally distributed dataset. The outliers are represented individually by black crosses. For this comparison, the IASI data are collocated at ±4° around the site location. The mean bias, the normalized mean bias (in parentheses) and the correlation coefficient are given for both methods.

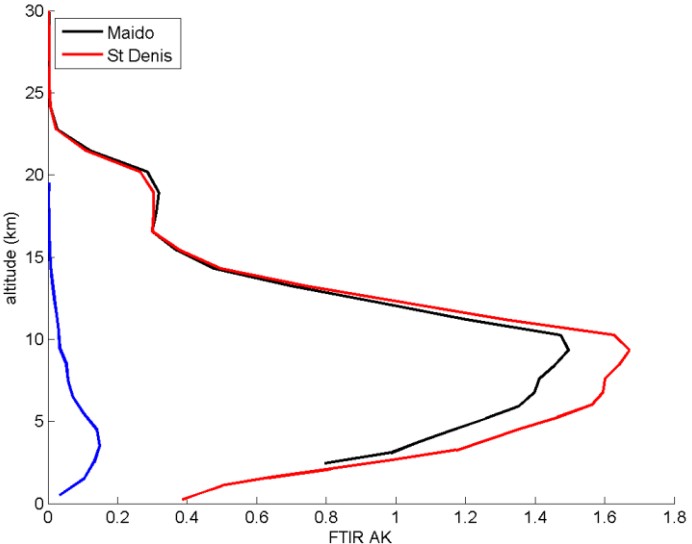

**Figure 11**. Mean total column AK for the FTIR ground-based measurements over Maido (black) and Saint-Denis (red) at La Réunion. Both stations are shown by green stars in Fig 1. Both FTIR stations have a degree of freedom for signal (DFS) close to 1. As reminder, the mean normalized Jacobians from Fig. 2 is plotted in blue.

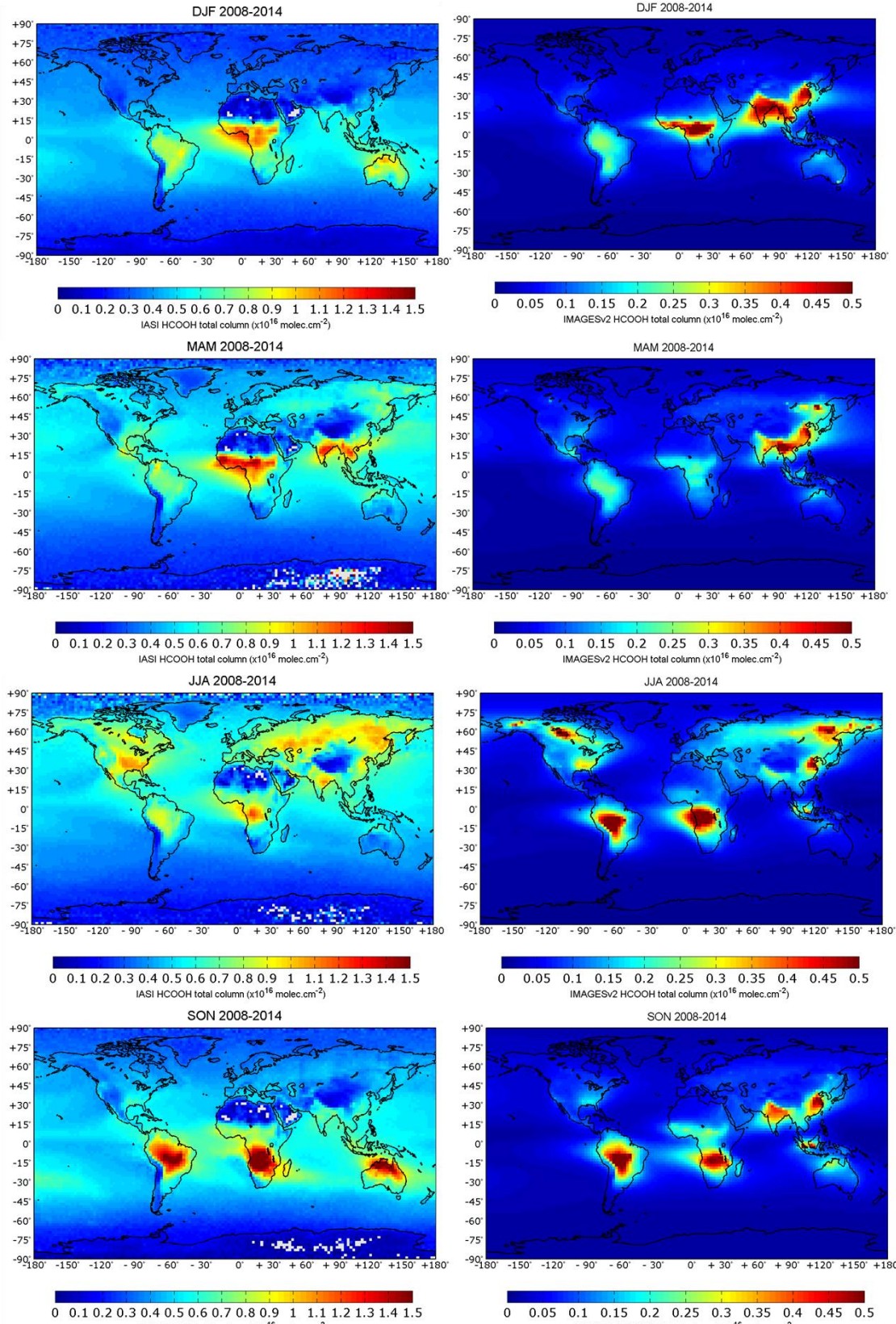

**Figure 12.** Seasonal HCOOH global distribution from 2008 to 2014 derived using the IASI radiance observations, gridded to IMAGESv2 horizontal resolution (2° lat × 2.5° lon) (left panel), and from IMAGESv2 (right panel). DJF=December-January-February, MAM=March-April-May, JJA=June-July-August, SON=September-October-November. Note the different color scale between both distributions.

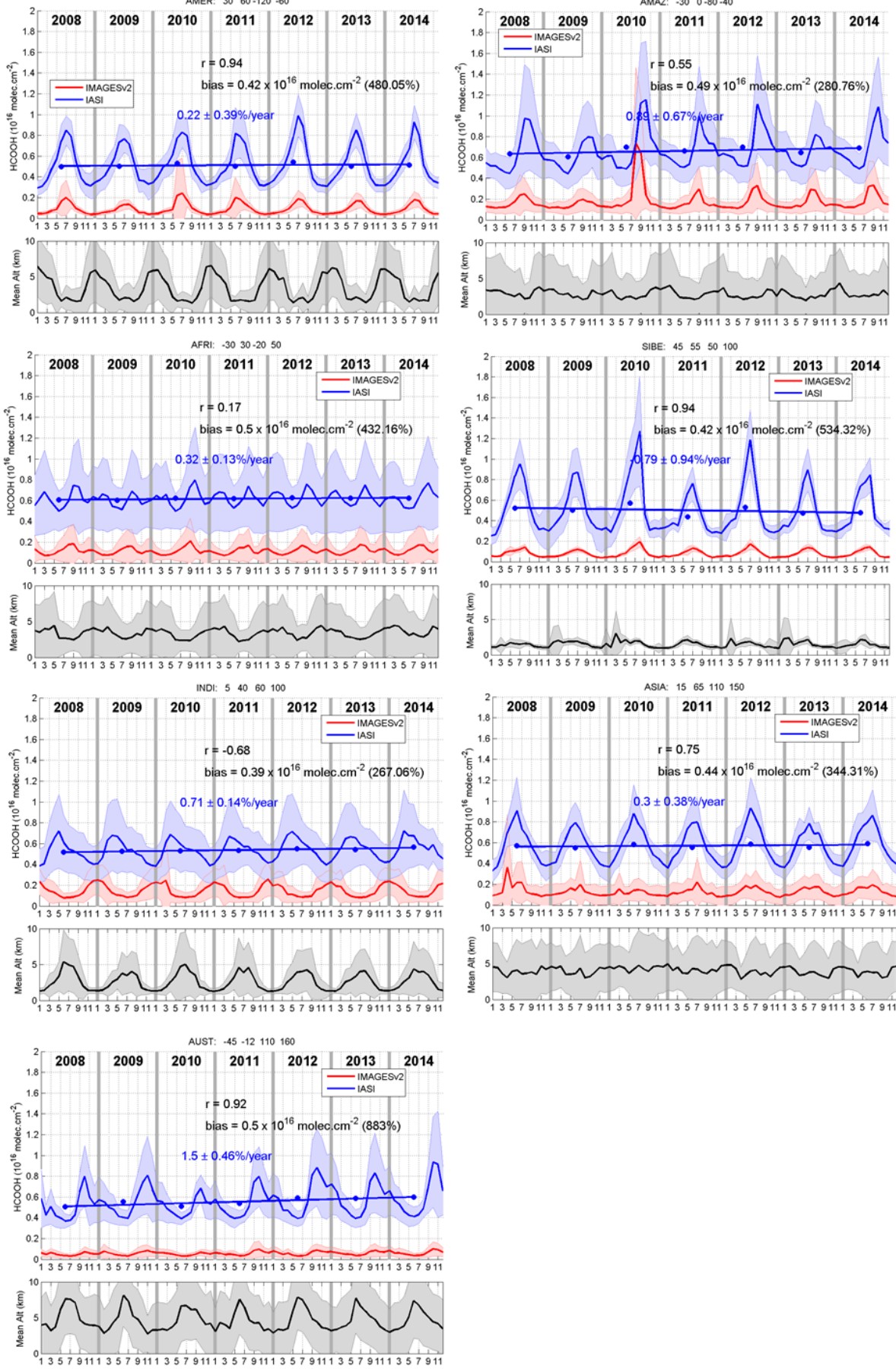

**Figure 13.** Time series of the monthly HCOOH column means for IASI (blue) and IMAGESv2 (red) over different regions highlighted by black boxes in Fig. 1, between 2008 and 2014. The coordinates of each box (latitude and longitude) are written on the top of each plot. The red and blue shaded areas correspond to the monthly standard deviation. The correlation coefficient, the mean bias and the normalized mean bias (in parentheses) for the full period are given on each plot. The blue dots correspond to the annual IASI mean. The linear regression on the annual IASI mean and the calculated linear trend are

800

also provided. On the bottom panel, the mean altitude of the maximum in the HCOOH vertical distribution from IMAGESv2 is plotted in black with the corresponding standard deviation represented by the black shade areas.