# Peer review of "HCOOH distributions from IASI for 2008-2014: comparison with ground-based FTIR measurements and a global chemistrytransport model"

_Atmospheric Chemistry and Physics, 2016_

## Referee Comment (RC1) · Anonymous Referee #2 · 28 Mar 2016

**Review: HCOOH distributions from IASI for 2008-2014: comparison with ground-based FTIR measurements and a global-chemistry transport model**

This paper presents a (very brief) description of an improved HCOOH from IASI retrieval method, evaluates the new method performance against the prior (Razavi et al., 2011) method, and carries out validations against both in situ FTIR HCOOH measurements and CTM (Imagesv2 ) output. There are numerous interesting results, however some topics need to be addressed more fully. Furthermore, while the paper is well organized over all, the paragraphs are often not well connected and it does not read very smoothly. I believe the paper should be published in ACP after some moderate revisions.

1.  Comments on content
    a.  Section 2.2:
        i.  This section could use some more details on which forward model and which retrieval code were used for the OEM retrievals. Are these the same as in Razavi et al., 2011(hereafter just Razavi)? This is not stated. Even if they are, providing this information adds clarity to the paper.
        ii.  The authors state that they used a large variability in the retrieval (350%) based on the retrieval settings of Razavi. I think a better term would be a loose constraint. I also do not understand what is meant by "based on the settings".
    b.  Section 2.3.1
        i.  Is the thermal contrast really defined as the difference between the surface temperature and the air temperature right above the surface? A more appropriate variable for satellite IR sensors is the difference between the surface temperature and the temperature at the peak of the instrument sensitivity. If the $\Delta BT$ from forward model runs is plotted as a function of thermal contrast (from the definition used in this paper), it will not be zero when the thermal contrast is zero. I suspect that the correction for $\Delta BT$ developed in this section would not be necessary if the definition I suggested were adopted. I would like to see a plot of $\Delta BT$ vs thermal contrast for various profiles using both definitions. If the authors feel this does not belong in the paper (though I believe it is an important point) they can submit the plots in their response. However, if the plots confirm my hypothesis I leave it to the editor to decide if this section should be omitted and the rest of the analysis redone.
        ii.  The last two sentences in this section appear contradictory. If rejecting negative values introduces a bias, then why are you rejecting negative values?
    c.  Section 2.3.2
        i.  The final paragraph in this section is very interesting. My experience with retrievals is that it is the lack of sensitivity to small changes in

background amounts that leads to the very large errors on these values. It would be very useful to users of this data if the authors could provide an estimate of the algorithm sensitivity, i.e., what is the threshold detection value and how it varies with thermal contrast.

d. Section 3.1

    i. The authors use the result from section 2.3.2 (high errors on low amounts, lower errors on high amounts), to justify their lower results compare to Razavi. However, Razavi found a similar pattern in their data analysis, so I do not believe this is the correct explanation, or at least not the entire source of the lower values, which actually occur nearly everywhere. The authors should comment on why their results are in general significantly lower over most regions/periods with enhanced HCOOH.

    ii. The long list of features evident in Figure 8 should be written less like a list. The numbering of each discussion point is useful, but I think the points could be expanded on and better connected.

    iii. The discussion on the Asian outflow is weak and unclear, as the IASI total columns are not provided for the PEM campaign periods/regions.

    iv. A possible reason for the high values over India should be provided.

e. Section 3.2

    i. This section requires at least some description of the FTIR: spectral resolution, noise level, sensitivity.

    ii. The paragraph starting at line 240 is especially confusing, as the OEM results are not shown. A plot or table would be helpful.

    iii. Why does the FTIR AK peak higher and have a broader peak?

    iv. Where does equation 3 come from?

2. Minor changes

a. Introduction

Line 24: …dependence on thermal contrast is taken into account…

Line 31: …highlights the difficulty of retrieving total columns from IASI measurements over mountainous regions…

Line 48: …, and to a lesser extent through oxidation by the OH radical.

Line 61: … the existence of unknown direct fluxes of HCOOH.

Line 63: Nadir looking atmospheric sensors can derive global distributions of trace gases, but with less vertical sensitivity than airborne or some ground-based measurements, such as FTIR instruments. Their extended spatial coverage allows for observations over remote regions ….

Line 69: Are the ACE data also monthly?

Line 70: … a low radiometric and high spatial coverage. HCOOH is a weak absorber, so it is challenging to …

Line 75: Which discrepancies? Please elaborate.

Line 77: …, suitable for both enhanced and background ….

b. Section 2

Line 87: …October 2006 and has provided more than eight years …

Line 88: …September 2012. Owing to their wide swath each instrument …

Line 93: Suggested rewriting of this paragraph:

Analysis of the mean of the normalized (by what??) Jacobians (Fig. 2) over the spectral range used by IASI for the HCOOH retrievals 1095-1114 cm-1(Is this correct? Later in the text the authors state they use the channels at 1103, 1105 and 11909 cm-1.) for a set of representative geographical regions (see Fig. 1 and next section) shows that IASI is sensitive to tropospheric HCOOH signal between 1 and 6 km.

Line 103: …columns of HCOOH using a set of conversion factors derived from OEM retrievals.

Line 127: …remote areas

Line 156: …of the method are …

Line 156: … and the lack of an error budget.

Line 159: … OEM retrievals. To provide an estimate of the algorithm error simulations were performed …

Line 170: …, Razavi et al. (2011), who find a mean RD …

c.  Section 3

Line 190: …Equator, with the highest values between 0-10°N, but with large variability, as the maximum was $3.5 \times 10^{16}$ molec/cm$^2$, but the monthly mean in this region was only $0.5 \times 10^{16}$ molec/cm$^2$.

Line 192-231: As noted above, please make this section less list-like. Some specific changes:

Line 192:  A number of features are evident and are discussed below:

(1) A particularly striking feature are the large hotspots over Russia …

(7) These states are flagged as biogenic emission regions …

---

## Referee Comment (RC2) · Anonymous Referee #1 · 4 Apr 2016

This paper presents results from an improved retrieval method for obtaining total columns of formic acid globally during seven years from IASI radiance spectra. This simple and computational inexpensive method has been used before for other species and presents some improvements to results obtained before for HCOOH by the same group. It is based on the calculation of conversion factors starting from a representative set of formal retrievals with the optimal estimation method, and using these to convert brightness temperature differences to column amounts. The results presented show improvements to previous attempts to derive reliable global distributions of HCOOH. This work has the potential to be published in ACP after a significant improvement of the manuscript is carried out by the authors.

In particular, I would like to point out that it is very poorly written in terms of wording and sentence structures, which makes the text at times very difficult to read and follow. Some (but clearly not all) minor corrections are listed below. It is important that the text is revised and improved by someone with good English skills. Also, some sections could be shortened and the key points could be better explained in a more concise manner without leaving important information out. Additionally to this, please consider the following points for improving the content and structure of the manuscript:

l.61 why would this be necessarily from a direct flux of HCOOH, couldn't it be also secondary formation from other unknown VOCs?

l94. Please comment on how (and why) the Averaging Kernels of the ground-based FTIR retrievals can be compared to the normalized Jacobians from the IASI retrievals. I could't find any information on this also in section 3.2

Fig 2. This figure would be more appropriate later on in the section describing the comparison with FTIR. Also, separate into two adjacent plots with common y-axis and individual x-axis for both the normalized Jacobians (left) and FTIR AK (right). Avoid the inset and use larger labels if it is to fit into a one column of the text.

Fig 2 caption. What is a "degree of freedom of signal"?? Ground-based FTIR retrievals often report the degrees of freedom (DOF) with respect to the independent layers sensed. A value of 1 would mean that no information in the vertical distribution is accessible. Does DFS refer to the same thing?

l.101 use the more conventional expression with B as subscript $\Delta T\_B$ here and throughout the manuscript.

l.102 I think the use of spectral microwindows is here more appropriate than "spectral channels".

l.110 what are these "mean RMS" differences? Do you refer to residuals? Please be clearer.

l.117 again, unclear of how you define channels. Are these mean brightness temperatures within a spectral range (microwindow) or rather just a value at a specific wavenumber?

l.123 Why not use tao for termal contrast as in previous studies?

Fig 3. Labels are missing in the plot to the right, if same as the one to the left, just include it to the x-axis. Also for the color palette. Use a) and b) to describe the plots as in Fig 4.

l.125 The linear regression for obtaining the conversion factors from the correlation between the OEM method and the $\Delta T\_B$'s was gathered from retrievals performed in different areas of the world, representing different conditions, which is good. The question is if these areas are treated separately would result in very unique conversions factors (something not shown in Fig. 3) and which could be used to improve the conversion from $\Delta T\_B$ to total columns. Is it sufficient to consider a correction for the dependance to the thermal contrast? Please comment.

Fig 4b. Use same scales in the x and y-axis as in Fig 3a.

l.152. What do you mean with only negative averages are being filtered out? You have just stated that negative values would produce a bias, so no filtering should be performed. Do you refer to columns used for the comparisons with FTIR and modeled data?

l.178 The results part showing the description and interpretation of the obtained global distributions should go AFTER the comparison with ground-based FTIR measurements and the model results. Maybe as section 4: Results. It is important to know how reliable (or not) the data are before using them for interpretation.

l.276 If the large biases found between the retrieved columns and this work does not come from using a simplified retrieval method as opposed to the OEM, then explain where the bias comes from. The explanation that IASI overestimates for background

levels in La Reunion because of the larger errors in the conversion from brightness temperatures is not valid in the case of Wollongong, or is it?. Please provide with a more solid explanation.

l.255 The altitude correction performed to both FTIR and IASI total columns is poorly argumented. Despite the fact that the correlation might improve, it may do so for the wrong reasons. I don't agree the authors should do this correction. It is quite feasible that a mountain site might not be sensing a plume further down while the broader pixel size of the IASI instrument covering lower altitudes might very well be detecting it.

l.260 If daily averages from FTIR are used instead of a more constrained time with respect to the IASI overpass time, then the authors should present the results showing that there is no improvement. I don't understand why they say the correlation does increase when using a +/-2h criterion and still don't apply it. In my opinion for a compound with such a short lifetime, in the range of hours, a more constrained time criterion than daily averages should be used in this study.

l.288 The authors decide to use a broad spatial coincidence criterion when comparing IASI with FTIR measurements, broader than in previous studies. Please provide with a more thorough explanation of why this decision was made. There seems to be enough IASI measurements to still have enough coincidences.

Minor corrections

l.20 "There are, however, large uncertainties on the sources and sinks of HCOOH and is therefore misrepresented . . ."

l.24 "The dependence. . ." sentence unclear

l.40 rewrite "is among the most. . ."

l.49 rewrite to make a clear and correct sentence

l.54 rewrite "despite the. . ."

l.58 rewrite "of emissions. . ."

l.67 rewrite "provided. . ." since that instrument is no longer operational

l.68 missing argument, ACE provides what in the upper troposphere and how often?

l.71 rewrite ", so it is a challenge to . . . radiances."

l.74 rewrite "during the summer. . ."

l.74-82 sentences poorly written, rewrite.

l.76 which method?

l.77 rewrite "over the period 2008-2014"

l.86 rewrite ". . . Fourier transform infrared spectrometer."

l.87 rewrite "on board"

l.101. why would you use the word robust if the drawback of this method, apart of being computationally cheap, has large errors and no AKs (see l.156).

l.111 "This" refers to what? Use proper sentence structures.

l.112 redundant use of "conversion" in one sentence

l.202-211 This paragraph should probably go a the end of the listing 1-10 as it refers to something general and not a specific region.

l.243. remove "way"

l.266 "strict of stricter"?
* * *

---

## Author Comment (AC1) · 6 Jun 2016

**Reply to Anonymous Referee #2**

This paper presents a (very brief) description of an improved HCOOH from IASI retrieval method, evaluates the new method performance against the prior (Razavi et al., 2011) method, and carries out validations against both in situ FTIR HCOOH measurements and CTM (Imagesv2 ) output. There are numerous interesting results, however some topics need to be addressed more fully. Furthermore, while the paper is well organized over all, the paragraphs are often not well connected and it does not read very smoothly. I believe the paper should be published in ACP after some moderate revisions.

The authors would like to thank the Referee for his careful reading of the manuscript and for his constructive comments. This paper certainly benefited from these corrections. A detailed point by point reply (in blue) is provided hereafter.

1.Comments on content

a.Section 2.2:

i.This section could use some more details on which forward model and which retrieval code were used for the OEM retrievals. Are these the same as in Razavi et al., 2011 (hereafter just Razavi)? This is not stated. Even if they are, providing this information adds clarity to the paper.

We thank the referee for this comment. We added the following sentence (in bold):

"In the current work, the main difference with the previous IASI HCOOH determination in Razavi et al. (2011) is the use of retrieved total columns over selected regions to determine conversion factors, instead of the use of forward simulations. **The OEM implemented in the line-by-line radiative transfer model Atmosphit (Coheur et al., 2005) has been used as in Razavi et al. (2011)."**

With the following reference:

Coheur, P.-F., Barret, B., Turquety, S., Hurtmans, D., Hadji-Lazaro, J., and Clerbaux, C.: Retrieval and characterization of ozone vertical profiles from a thermal infrared nadir sounder, J. Geophys. Res., 110, D24303, doi:10.1029/2005JD005845, 2005.

ii. The authors state that they used a large variability in the retrieval (350%) based on the retrieval settings of Razavi. I think a better term would be a loose constraint. I also do not understand what is meant by "based on the settings".

We agree with the referee that the term "based on the settings" is confusing, so it was deleted. We also changed the "large variability" by "loose constraint".

b.Section 2.3.1

i. Is the thermal contrast really defined as the difference between the surface temperature and the air temperature right above the surface? A more appropriate variable for satellite IR sensors is the difference between the surface temperature and the temperature at the peak of the instrument sensitivity.

If the $\Delta BT$ from forward model runs is plotted as a function of thermal contrast (from the definition used in this paper), it will not be zero when the thermal contrast is zero. I suspect that the correction for $\Delta BT$ developed in this section would not be necessary if the definition I suggested were adopted.

I would like to see a plot of $\Delta BT$ vs thermal contrast for various profiles using both definitions. If the authors feel this does not belong in the paper (though I believe it is an

important point) they can submit the plots in their response. However, if the plots confirm my hypothesis I leave it to the editor to decide if this section should be omitted and the rest of the analysis redone.

The definition of TC is similar to that which was used for several previous papers ((e.g. Clerbaux et al. (2009) – cited in the manuscript; or Bauduin et al. (2014)). The correction on $\Delta T_b$ is also similar to the correction performed in previous studies (e.g. Razavi et al. (2011)).

S. Bauduin, L. Clarisse, C. Clerbaux, D. Hurtmans, and P.-F. Coheur: IASI observations of sulfur dioxide ($SO_2$) in the boundary layer of Norilsk , J. Geophys. Res., 119, 4253–4263, doi:10.1002/2013JD021405, 2014.

As can be seen on the Jacobians plotted in Fig.2, the maximum of vertical sensitivity of IASI is located at around 3 km. As suggested, the same test as performed in Fig.4a was done by perturbing the temperature profile at 3 km, by ±5K.

A "thermal contrast (TC)" defined here as the difference between the surface temperature and the temperature at 3km is calculated.

A similar but slightly lower correlation (see figure below) is found with the new calculation than the result found in the paper (r=0.58 hereafter, and r=0.61 in the paper). The equation found is also similar.

[Figure]

Fig. Scatter plot between the simulated $\Delta T_b$ and the TC=Tsurf-T3km, for one fixed HCOOH total column ($0.6 \times 10^{16}$ molec/cm$^2$).

ii. The last two sentences in this section appear contradictory. If rejecting negative values introduces a bias, then why are you rejecting negative values?

We wanted to say that the negative values were also kept to calculate the averages, but if these averages were found to be negative, then they were filtered out. We agree the sentences were confusing.

Now the new sentence (in bold) reads:

"Note that this conversion could lead to negative total columns. If we eliminated all the negative values and kept only all the positive values, we would introduce an artificial bias in the average. **For comparisons with zonal or temporal averages, the negative total columns were included in the average. But when the average was found to be negative, it was filtered out.**"

c.Section 2.3.2

i.The final paragraph in this section is very interesting. My experience with retrievals is that it is the lack of sensitivity to small changes in background amounts that leads to the very large errors on these values. It would be very useful to users of this data if the authors could provide an estimate of the algorithm sensitivity, i.e., what is the threshold detection value and how it varies with thermal contrast.

It is an interesting comment. We added this following information:

"Considering the detection threshold defined as $2\sigma$ on the $\Delta T_b$ (=0.30K), an indicative total column detection threshold was calculated using our conversion factors. To do so, forward simulations were performed for different total columns and TC. The result is illustrated in Fig. 6b and this shows that for the less favorable TC condition (TC=0K) the detection limit of HCOOH is close to $0.6\times10^{16}$ molec/cm$^2$ ($0.4\times10^{16}$ molec/cm$^2$ for $\sigma$). This detection limit is improved with higher TC."

With this corresponding figure:

[Figure]

**Figure 6.** a: Variation of the mean relative difference between the total columns derived from the $\Delta T_b$ conversion and the a priori total columns (used as input in the forward simulations) according to the a priori used. The black solid line corresponds to a relative difference equal to 0 and the dashed black lines to $\pm40\%$. b: Variation of the simulated $\Delta T_b$ for different HCOOH total columns and TC. The dashed black lines correspond to a $\Delta T_b$ equal to 0.15K, 0.30K and 0.45K. 0.15 K corresponds to the IASI radiometric noise in the HCOOH spectral range (see Section 2.1).

d. Section 3.1

i.The authors use the result from section 2.3.2 (high errors on low amounts, lower errors on high amounts), to justify their lower results compare to Razavi. However, Razavi found a similar pattern in their data analysis, so I do not believe this is the correct explanation, or at least not the entire source of the lower values, which actually occur nearly everywhere. The authors should comment on why their results are in general significantly lower over most regions/periods with enhanced HCOOH.

It is a very good comment. That is why we added an additional explanation with this following sentence:

"It is also important to note that in Razavi et al. (2011), only averaged data in a 0.5°×0.5° grid with TC higher than 5K were considered. This implies that only data with a strong signal were used, probably overestimating the threshold of the $\Delta T_b$ and thus also the retrieved columns."

ii. The long list of features evident in Figure 8 should be written less like a list. The numbering of each discussion point is useful, but I think the points could be expanded on and better connected.

This part (lines 192-231 of the ACPD version) was rewritten and the list is now removed (see the last comment).

iii. The discussion on the Asian outflow is weak and unclear, as the IASI total columns are not provided for the PEM campaign periods/regions.

To clarify this point, we added the following details (in bold in the sentences):

[revised manuscript text omitted]

ii. The paragraph starting at line 240 is especially confusing, as the OEM results are not shown. A plot or table would be helpful.
These details (below) are now available in a supplement.

**Table S1** Mean bias in $10^{16}$ molec/cm$^2$, normalized mean bias in percent (in parentheses), between the daily FTIR measurements and the daily IASI co-located mean data, retrieved by OEM and between the daily FTIR measurements and the columns using the conversion factors. The IASI data were collocated within 0.5° of the site location. The number of coincidence days is given for each site. For this test, only the five first days of each month in 2009 were retrieved as done in the Section 2.3.2 retrieval approach.

|  | Saint-Denis (12 days) | Wollongong (13 days) | Jungfraujoch (10 days) |
|---|---|---|---|
| IASI column, OEM | 0.31 (110.65%) | -0.04 (-13.11%) | -0.86 (-73.96%) |
| IASI column, converted | 0.26 (91.72%) | -0.05 (-15.14%) | -0.8 (-69.24%) |

[Figure]

**Figure S2.** Scatterplot between the HCOOH columns retrieved by OEM (HCOOH OEM) and the corresponding columns calculated by the conversion factors (HCOOH conv). The IASI spectra were selected at ±0.5° around each site location (JUN = Jungfraujoch, WOL = Wollongong, SDE = Saint-Denis, MAI = Maido). The mean bias, the normalized mean bias, and the correlation coefficient are reported. The blue line corresponds to the linear regression and the corresponding equation is also provided. For this test, only the five first days of each month in 2009 were retrieved.

And we now refer to these results as (in bold):
"The current IASI retrieved columns were also compared with a set of columns retrieved by OEM around the sites. For each OEM-based retrieved column, the corresponding column using the conversion factors was calculated, showing that the current dataset and the OEM-based retrieval are in agreement (correlation ranging from 0.7 to 0.8, with an underestimation of the columns calculated with the conversion factors between 6 and 15%) **(Fig. S2).** It is also worth noting that similar biases were found between the columns retrieved by OEM around the ground-based locations and the FTIR columns as between the columns retrieved in this work and the FTIR ones **(Tab. S1).**"

iii. Why does the FTIR AK peak higher and have a broader peak?
It is normal to not have the same peak between a ground-based FTIR and a satellite FTIR. Both instruments are measuring in the IR but the spectral resolution is different and the geometry is different.
The DFS for the FTIR is close to 1, meaning that the signal corresponds to a column. Since 98% of the HCOOH is located below 20 km of altitude, it means that the FTIR mainly measured the column between the surface and 20 km.

iv. Where does equation 3 come from?

The columns are scaled with a height factor of 7.4. This equation is a simplified formula which is a variation of the hypsometric equation (Wallace and Hobbs, 1977).

Wallace, J. M. and Hobbs, P. V.: Atmospheric Science: An Introductory Survey, 1977.

You can find an example of its use in:

De Mazière, M. et al., Comparisons between SCIAMACHY Scientific Products and Ground-Based FTIR Data for Total Columns of CO, $CH_4$ and $N_2O$, Proceedings of the Second Workshop on the Atmospheric Chemistry Validation of ENVISAT (ACVE-2), ESA-ESRIN, Frascati, Italy, 3-7 May 2004 (ESA SP-562, August 2004) ESC02MDM.

We added the sentence to the paper "This simplified formula is a variation of the hypsometric equation (Wallace and Hobbs, 1977)" with the corresponding reference.

2. Minor changes
a. Introduction
Line 24: ...dependence on thermal contrast is taken into account...
Corrected

Line 31: ...highlights the difficulty of retrieving total columns from IASI measurements over mountainous regions...
Changed

Line 48: ..., and to a lesser extent through oxidation by the OH radical.
Done

Line 61: ... the existence of unknown direct fluxes of HCOOH.
Changed

Line 63: Nadir looking atmospheric sensors can derive global distributions of trace gases, but with less vertical sensitivity than airborne or some ground-based measurements, such as FTIR instruments. Their extended spatial coverage allows for observations over remote regions ....
We prefer to keep the previous version: "Nadir looking atmospheric sensors allow to derive…".

Line 69: Are the ACE data also monthly?
No, per season. We then added the information (in bold):
"…and the solar-occultation Atmospheric Chemistry Experiment (ACE) **provides seasonal global distribution** in the upper troposphere (e.g. González Abad, 2009)."

Line 70: ... a low radiometric and high spatial coverage. HCOOH is a weak absorber, so it is challenging to ...
We changed this by "HCOOH is a weak absorber, so it is a challenge to …"
This change was requested by Referee 1.

Line 75: Which discrepancies? Please elaborate.
An additional sentence was added:

"Indeed, the total columns from R'Honi et al. (2013) were on average a factor of 2 lower than in Razavi et al. (2011) (around a factor of 1.5 for columns higher than $5\times10^{16}$ molec.cm$^{-2}$ and 2.3 for columns lower than $5\times10^{16}$ molec.cm$^{-2}$)."

Line 77: ..., suitable for both enhanced and background ....
"large" was replaced by "enhanced" as requested.

b.Section 2
Line 87: ...October 2006 and has provided more than eight years ...
Changed

Line 88: ...September 2012. Owing to their wide swath each instrument ...
Changed. Now it reads: "Owing to its wide swath, each instrument delivers near global coverage twice per day at around 9:30 local time (AM and PM)"

Line 93: Suggested rewriting of this paragraph:  Analysis of the mean of the normalized (by what??) Jacobians (Fig. 2) over the spectral range used by IASI for the HCOOH retrievals 1095-1114cm-1(Is this correct ? Later in the text the authors state they use the channels at 1103, 1105 and 11909 cm-1.) for a set of representative geographical regions (see Fig. 1 and next section) shows that IASI is sensitive to tropospheric HCOOH signal between 1 and 6 km.
Yes, the retrieval was done for the spectral range between 1095 and 1104 cm$^{-1}$, but the calculation of $\Delta T_b$ used the channels at 1103, 1105 and 1109 cm$^{-1}$.
Normalized Jacobians means Jacobians integrated vertically.
The sentence has been changed as below:
"Analysis of the mean of the normalized Jacobians (Fig. 2) over the spectral range used by IASI for the HCOOH retrievals (1095-1114 cm$^{-1}$), for a set of representative geographical regions (see Fig. 1 and next section), shows that IASI is sensitive to tropospheric HCOOH between 1 and 6 km."

Line 103: ...columns of HCOOH using a set of conversion factors derived from OEM retrievals.
Now, the sentence is: "In a second step, the $\Delta T_b$ were converted into total columns of HCOOH using conversion factors derived from a set of data retrieved by OEM."

Line 127: ...remote areas
Corrected

Line 156: ...of the method are ...
It was corrected and it is "…the drawbacks of the method are.."

Line 156: ...and the lack of an error budget.
It was added

Line 159: ... OEM retrievals. To provide an estimate of the algorithm error simulations were performed ...
The sentence has been modified.

Line 170: ..., Razavi et al. (2011), who find a mean RD ...
Changed. We wrote "Razavi et al. (2011), who found a mean RD…"

c. Section 3

Line 190: ...Equator, with the highest values between 0-10°N, but with large variability, as the maximum was $3.5\times10^{16}$molec/cm$^2$, but the monthly mean in this region was only $0.5\times10^{16}$molec/cm$^2$.

The sentence has been modified. Now it reads:

"They showed a gradient of columns from the Poles to the Equator, with the highest values between 0 and 10°N, but with large variability, as the maximum was $3.5\times10^{16}$ molec/cm$^2$, but the monthly mean in this region was only $0.5\times10^{16}$ molec/cm$^2$".

Line 192-231: As noted above, please make this section less list-like. Some specific changes:
Line 192: A number of features are evident and are discussed below:
(1) A particularly striking feature are the large hotspots over Russia ...
(7) These states are flagged as biogenic emission regions…
The paragraph now reads:

[revised manuscript text omitted]

---

## Author Comment (AC2) · 6 Jun 2016

**Reply to Anonymous Referee #1**

The authors thank the Referee for his careful reading of the manuscript and for his thorough review. A detailed point by point reply (in blue) is provided hereafter.

This paper presents results from an improved retrieval method for obtaining total columns of formic acid globally during seven years from IASI radiance spectra. This simple and computational inexpensive method has been used before for other species and presents some improvements to results obtained before for HCOOH by the same group. It is based on the calculation of conversion factors starting from a representative set of formal retrievals with the optimal estimation method, and using these to convert brightness temperature differences to column amounts. The results presented show improvements to previous attempts to derive reliable global distributions of HCOOH. This work has the potential to be published in ACP after a significant improvement of the manuscript is carried out by the authors.

In particular, I would like to point out that it is very poorly written in terms of wording and sentence structures, which makes the text at times very difficult to read and follow. Some (but clearly not all) minor corrections are listed below. It is important that the text is revised and improved by someone with good English skills. Also, some sections could be shortened and the key points could be better explained in a more concise manner without leaving important information out. Additionally to this, please consider the following points for improving the content and structure of the manuscript:

The text was revised in order to improve the English.

l.61 why would this be necessarily from a direct flux of HCOOH, couldn't it be also secondary formation from other unknown VOCs?
This comes from the conclusions given by Millet et al. (2015), -see reference in the manuscript:
In this paper it is said that "This indicates one or more large missing HCOOH sources, and suggests either a key gap in current understanding of hydrocarbon oxidation or a large, unidentified, direct flux of HCOOH."
Moreover, the secondary formation from other unknown VOCs is included in "hydrocarbon oxidation".

l.94. Please comment on how (and why) the Averaging Kernels of the ground-based FTIR retrievals can be compared to the normalized Jacobians from the IASI retrievals. I could't find any information on this also in section 3.2
We now provide this information in the section 3.2:
"The AKs indicate the vertical sensitivity of the retrieval. The Jacobians express the sensitivity of the radiative transfer model and the IASI instrument (through its instrumental function) to the variation of HCOOH in the atmosphere. Both functions then give a good indication of the vertical sensitivity for each data set."

Fig 2. This figure would be more appropriate later on in the section describing the comparison with FTIR. Also, separate into two adjacent plots with common y-axis and individual x-axis for both the normalized Jacobians (left) and FTIR AK (right). Avoid the inset and use larger labels if it is to fit into a one column of the text.

As suggested by the reviewer, we enlarged the labels for the Jacobians plot and the inset is removed. We also decided to separate both plots (Jacobians and averaging kernels) as below. Now there are 2 figures:

[Figure]

**Figure 2.** Mean normalized Jacobians of all retrieved spectra (over the 7 selected regions) as a function of altitude.

[Figure]

**Figure 11**. Mean total column AK for the FTIR ground-based measurements over Maido (black) and Saint-Denis (red) at La Réunion. Both stations are shown by green stars in Fig 1. Both FTIR stations have a degree of freedom for signal (DFS) close to 1. As reminder, the mean normalized Jacobians from Fig. 2 is plotted in blue.

This Fig 11 is located in the Section 3.2 "Comparison with ground-based FTIR measurements"

Fig 2 caption. What is a "degree of freedom of signal"?? Ground-based FTIR retrievals often report the degrees of freedom (DOF) with respect to the independent layers sensed. A value of 1 would mean that no information in the vertical distribution is accessible. Does DFS refer to the same thing?

Thanks for noting this: we change the 'of' into 'for' so that DFS = degree of freedom for signal. Yes the concept is similar for ground-based instrument and satellite retrieved data and both acronyms DOFs and DFS are used in the literature, e.g. for DFS:

Deeter, M. N., H. M. Worden, D. P. Edwards, J. C. Gille, and A. E. Andrews (2012), Evaluation of MOPITT retrievals of lower-tropospheric carbon monoxide over the United States, J. Geophys. Res., 117, D13306, doi:10.1029/2012JD017553.

Worden, H. M., Deeter, M. N., Frankenberg, C., George, M., Nichitiu, F., Worden, J., Aben, I., Bowman, K. W., Clerbaux, C., Coheur, P. F., de Laat, A. T. J., Detweiler, R., Drummond, J. R., Edwards, D. P., Gille, J. C., Hurtmans, D., Luo, M., Martínez-Alonso, S., Massie, S., Pfister, G., and Warner, J. X.: Decadal record of satellite carbon monoxide observations, Atmos. Chem. Phys., 13, 837-850, doi:10.5194/acp-13-837-2013, 2013.

A DFS close to 1 means that the signal corresponds to a column, with no profile information available.

l.101 use the more conventional expression with B as subscript $\Delta T\_B$ here and throughout the manuscript.
As suggested, we changed it to $\Delta T_b$ in the text, in the captions and on the axes for the Figs. 3 & 4.

l.102 I think the use of spectral microwindows is here more appropriate than "spectral channels".
"Spectral channels" is the appropriate term for this study as the calculation of $\Delta T_b$ is based on specific channels and not on a full microwindow.

l.110 what are these "mean RMS" differences? Do you refer to residuals? Please be clearer.
The RMS is the square-root of the differences between the observed and the fitted spectra. The RMS mentioned in the manuscript is the mean of all RMS, thus the mean of all square-root of the residuals.
We agree, the word "difference" was forgotten in the text and it was confusing. We have deleted it.

l.117 again, unclear of how you define channels. Are these mean brightness temperatures within a spectral range (microwindow) or rather just a value at a specific wavenumber?
We used the value at a specific wavenumber:
Tb HCOOH at 1105 $cm^{-1}$
Tb ref 1 at 1103 $cm^{-1}$
Tb ref 2 at 1109 $cm^{-1}$
Thus the sentence "The reference channels used for the calculation of $\Delta T_b$ were chosen on both sides of the HCOOH channel (1105 $cm^{-1}$), i.e. at 1103.0 and 1109.0 $cm^{-1}$." is correct.

l.123 Why not use tao for termal contrast as in previous studies?
This definition of thermal contrast was used in several previous studies from our group (e.g. Clerbaux et al. (2009) – cited in the manuscript). It is a standard definition.

Fig 3. Labels are missing in the plot to the right, if same as the one to the left, just include it to the x-axis. Also for the color palette. Use a) and b) to describe the plots as in Fig 4.
Done

l.125 The linear regression for obtaining the conversion factors from the correlation between the OEM method and the $\Delta T\_B$'s was gathered from retrievals performed in different areas of the world, representing different conditions, which is good. The question is if these areas are treated separately would result in very unique conversions factors (something not shown in Fig. 3) and which could be used to improve the conversion from $\Delta T\_B$ to total columns. Is it sufficient to consider a correction for the dependance to the thermal contrast? Please comment.

Let's start the answer by a clarification.

From line 125, we described the correlation between the columns retrieved by OEM and the $\Delta T_b$. We highlighted the difficulty to convert these $\Delta T_b$ to columns using these coefficients since there is still an impact of the thermal contrast. Hence the conversion factors were found in section 2.3.1 and illustrated by Fig.4.

Indeed the factors will change with the used a priori columns. Since the idea was to have common coefficients for the full globe, all spectrum were gathered in a single set.

About Fig.3: If we separate different areas, we will obtain different correlations in each region, as it is characterized by different amount of HCOOH, temperature profile, etc.

About the dependence to the thermal contrast, this is a good remark. Razavi et al. (2011) took also into account the impact of $H_2O$ in their conversion. It is not presented in the paper, but we also checked if the $\Delta T_b$ were correlated to the $H_2O$ columns.

Hereafter you can see the scatterplot between the $\Delta T_b$ of the spectra used for the OEM-based retrieval over the seven regions and their $H_2O$ column. No clear correlation was found (r=0.01).

[Figure]

We also performed a similar test than the test performed for the TC. We modified the $H_2O$ profile by +10% and +20%. Thus 3 different profiles were used: $H_2O$ ref, $H_2O$ ref +10%, $H_2O$ ref + 20%. We performed forward simulations and we checked the correlation between the simulated $\Delta T_b$ and the $H_2O$ columns. We obtained the following scatterplot:

[Figure]

Fig 4b. Use same scales in the x and y-axis as in Fig 3a.
Done

l.152. What do you mean with only negative averages are being filtered out? You have just stated that negative values would produce a bias, so no filtering should be performed. Do you refer to columns used for the comparisons with FTIR and modeled data?
We wanted to say that the negative values were also kept to calculate the averages, but if these averages were found to be negative, then they were filtered out. We agree the sentences were confusing.
Now the sentence reads:
"For comparisons with zonal or temporal averages, the negative total columns were included in the average. But when the average was found to be negative, it was filtered out."

l.178 The results part showing the description and interpretation of the obtained global distributions should go AFTER the comparison with ground-based FTIR measurements and the model results. Maybe as section 4: Results. It is important to know how reliable (or not) the data are before using them for interpretation.
This is an interesting remark but we presented this section as the first section in the analysis part since we compared the new dataset with the work done by Razavi et al. (2011). We decided to keep this structure.
The Fig. 7 highlights the difference on the global distribution between both retrievals and it is important to analyze the reasons for the differences before further interpretation of the data.

To clarify our analysis, we also added this information at the beginning of the section (in bold):
"Mean HCOOH global distributions (averaged on a 0.5°×0.5° grid) from IASI for the 2008-2014 period are presented in Fig. 7 and compared with columns obtained using the retrieval method of Razavi et al. (2011). Note that Razavi et al. (2011) retrieved only total columns over land. Except over Indonesia, lower values are observed over the source regions with the updated dataset. The previous section shows that large positive RDs are expected for very low true columns. Even if the columns from Razavi et al. (2011) are not the true columns, this could explain why the total columns for this study are higher over remote areas (e.g. deserts) than those obtained using the methodology described by Razavi et al. (2011). **It is also**

**important to note that in Razavi et al. (2011), only averaged data in a 0.5°×0.5° grid with TC higher than 5K were considered. This implies that only data with a strong signal were used, probably overestimating the threshold of the $\Delta T_b$ and thus also the retrieved columns.**"

Moreover, the Figs. 8 and S1 help to interpret the peaks observed on the time-series in the following section, i.e. for the comparison with the FTIR measurements. Thus it is a good reason to present the global distributions before the comparisons with the FTIR and the CTM.

l.276 If the large biases found between the retrieved columns and this work does not come from using a simplified retrieval method as opposed to the OEM, then explain where the bias comes from. The explanation that IASI overestimates for background levels in La Reunion because of the larger errors in the conversion from brightness temperatures is not valid in the case of Wollongong, or is it?. Please provide with a more solid explanation.
The bias was not coming from our conversion method. The overestimation of the background levels over La Réunion and at Wollongong could be a result of a low detection limit due to a low local TC.

l.255 The altitude correction performed to both FTIR and IASI total columns is poorly argumented. Despite the fact that the correlation might improve, it may do so for the wrong reasons. I don't agree the authors should do this correction. It is quite feasible that a mountain site might not be sensing a plume further down while the broader pixel size of the IASI instrument covering lower altitudes might very well be detecting it.
You are right but if this correction is not applied, the comparison will be biased for the sites at high altitudes due to the absence of the lower levels. Indeed, over these high altitude stations the retrieved column are truncated since the lowest layer are not represented.

The equation is a simplified formula which is a variation of the hypsometric equation (Wallace and Hobbs, 1977).
Wallace, J. M. and Hobbs, P. V.: Atmospheric Science: An Introductory Survey, 1977.

You can find an example of its use in:
De Mazière, M. et al., Comparisons between SCIAMACHY Scientific Products and Ground-Based FTIR Data for Total Columns of CO, $CH_4$ and $N_2O$, Proceedings of the Second Workshop on the Atmospheric Chemistry Validation of ENVISAT (ACVE-2), ESA-ESRIN, Frascati, Italy, 3-7 May 2004 (ESA SP-562, August 2004) ESC02MDM.

We added this sentence to the paper "This simplified formula is a variation of the hypsometric equation (Wallace and Hobbs, 1977)" with the corresponding reference.

We show below the same comparison between IASI and FTIR data as presented in the manuscript for Jungfraujoch and Maido, without the altitude correction:

[Figure]

l.260 If daily averages from FTIR are used instead of a more constrained time with respect to the IASI overpass time, then the authors should present the results showing that there is no improvement. I don't understand why they say the correlation does increase when using a +/-2h criterion and still don't apply it. In my opinion for a compound with such a short lifetime, in the range of hours, a more constrained time criterion than daily averages should be used in this study.

We agree the sentence was confusing. We decided to add this sentence (in bold).

"A more stringent criterion of ±2h was tested but provided similar results, except over Maido where the correlation increased to 0.6 without improvement of the bias. **The advantage of this daily average is the possibility to derive the seasonal variation over each site**. Over all sites, the broad patterns of seasonal and inter-annual variations were similarly captured by IASI and the ground-based FTIR."

The reviewer can see an illustration of this sentence with this figure:

[Figure]

**Figure. Left:** Time series of HCOOH over Wollongong between 2008 and 2014 for IASI (blue) and the ground-based FTIR (red) measurements. The IASI data are collocated at ±0.5° and ±2h around each FTIR measurement. The correlation coefficient, the mean bias for all years is given in blue on each plot. The blue shade error bar corresponds to the standard deviation on the IASI daily means. **Right:** As left panel but the IASI curve corresponds to daily averages and the IASI data are collocated at ±0.5° around the site location. This plot is the time-series presented in the paper.

l.288 The authors decide to use a broad spatial coincidence criterion when comparing IASI with FTIR measurements, broader than in previous studies. Please provide with a more thorough explanation of why this decision was made. There seems to be enough IASI measurements to still have enough coincidences.

The idea was to compare this updated dataset with the work performed by Razavi et al. (2011).

Stavrakou et al. (2012) presented a comparison with the colocation criteria used in our work. The number of available data from Razavi et al. (2011) is also less important than the current study due to their stringent criteria in their conversion (averaged TC in a 0.5°x0.5° grid > 5K).

We clarified this point in the text:

"The FTIR measurements were also used to evaluate the current HCOOH columns with those using the conversion from Razavi et al. (2011) (Fig. 10). **The colocation criteria have been enlarged to ±4° as used in the evaluation shown in Stavrakou et al. (2012). The criterion was enlarged since the number of available data from Razavi et al. (2011) around the sites was less important than for the current dataset."**

The reviewer can find hereafter the time-series over La Réunion (Saint Denis and Maido) site using the colocation criteria used in the paper (daily averages, ±4°) and illustrating the lower number of data with the technique from Razavi et al. (2011) (in green) compared to our work (in blue).

[Figure]

Minor corrections

l.20 "There are, however, large uncertainties on the sources and sinks of HCOOH and is therefore misrepresented..."

Changed.

"There are, however, large uncertainties on the sources and sinks of HCOOH and therefore HCOOH is misrepresented by global chemistry-transport models".

l.24 "The dependence..." sentence unclear

The sentence has been modified as below:

"The dependence of the measured HCOOH signal to the thermal contrast is taken into account in the conversion method."

l.40 rewrite "is among the most ..."

done

l.49 rewrite to make a clear and correct sentence
The sentence is now:
"HCOOH is a short-lived species and its lifetime is mainly determined by the precipitation rate. The lifetime ranges between 2 days during the rainy season and 6 days in the dry season in the boundary layer (Sanhueza et al., 1996). The global lifetime in the troposphere is 3–4 days (Paulot et al., 2011; Stavrakou et al., 2012)."

l.54 rewrite "despite the..."
done

l.58 rewrite "of emissions..."
the "s" was added.

l.67 rewrite "provided..." since that instrument is no longer operational
corrected.

l.68 missing argument, ACE provides what in the upper troposphere and how often?
We added the information (in bold):
"…and the solar-occultation Atmospheric Chemistry Experiment (ACE) **provides seasonal global distribution** in the upper troposphere (e.g. González Abad, 2009)."

l.71 rewrite ", so it is a challenge to...radiances."
done

l.74 rewrite "during the summer..."
done

l.74-82 sentences poorly written, rewrite.
The sentences are now:
"These studies however highlighted discrepancies between the retrieved distributions and especially within enriched HCOOH air masses as, for instance, over large forest fires. Indeed, the total columns from R'Honi et al. (2013) were on average a factor of 2 lower than in Razavi et al. (2011) (around a factor of 1.5 for columns higher than $5 \times 10^{16}$ molec.cm$^{-2}$ and 2.3 for columns lower than $5 \times 10^{16}$ molec.cm$^{-2}$). In this paper, we present an update of the method used in Razavi et al. (2011), in order to derive HCOOH distributions over both land and sea, suitable for both enhanced and background concentrations over the period 2008-2014."

l.76 which method?
The information in bold is added:
"…an update of the method **used in Razavi et al. (2011),…"**

l.77 rewrite "over the period 2008-2014"
done

l.86 rewrite "...Fourier transform infrared spectrometer."
Done. Now the upper-case letters are deleted except on the word Fourier.

l.87 rewrite "on board"

done

l.101. why would you use the word robust if the drawback of this method, apart of being computationally cheap, has large errors and no AKs (see l.156).
The term "robust" was deleted.

l.111 "This" refers to what? Use proper sentence structures.
Now, it is "This RMS value".

l.112 redundant use of "conversion" in one sentence
It is changed. The sentence is now:
"The conversion factors allowing the calculation of total columns based on $\Delta T_b$ values…"

l.202-211 This paragraph should probably go a the end of the listing 1-10 as it refers to something general and not a specific region.
The initial order was chosen in order to separate the description of the annual distributions (Fig.8) and the monthly distributions (Fig.S1).
As requested by the other reviewer, this part was rewritten. We also moved the mentioned paragraph at the end of the section.

l.243. remove "way"
done

l.266 "strict of stricter"?
Thank you for finding this typing error. It is strict.